# Random Coordinate Langevin Monte Carlo

## Abstract

Langevin Monte Carlo (LMC) is a popular Markov chain Monte Carlo sampling method. One drawback is that it requires the computation of the full gradient at each iteration, an expensive operation if the dimension of the problem is high. We propose a new sampling method: Random Coordinate LMC (RC-LMC). At each iteration, a single coordinate is randomly selected to be updated by a multiple of the partial derivative along this direction plus noise, and all other coordinates remain untouched. We investigate the total complexity of RC-LMC and compare it with the classical LMC for log-concave probability distributions. When the gradient of the log-density is Lipschitz, RC-LMC is less expensive than the classical LMC if the log-density is highly skewed for high dimensional problems, and when both the gradient and the Hessian of the log-density are Lipschitz, RC-LMC is always cheaper than the classical LMC, by a factor proportional to the square root of the problem dimension. In the latter case, our estimate of complexity is sharp with respect to the dimension.

## 1 Introduction

Monte Carlo sampling plays an important role in machine learning (Andrieu et al., 2003) and Bayesian statistics. In applications, the need for sampling is found in atmospheric science (Fabian, 1981), epidemiology (Li et al., 2020), petroleum engineering (Nagarajan et al., 2007), in the form of data assimilation (Reich, 2011), volume computation (Vempala, 2010) and bandit optimization (Russo et al., 2018).

In many of these applications, the dimension of the problem is extremely high. For example, for weather prediction, one measures the current state temperature and moisture level, to infer the flow in the air, before running the Navier–Stokes equations into the near future (Evensen, 2009). In a global numerical weather prediction model, the degrees of freedom in the air flow can be as high as $10^9$. Another example is from epidemiology: When a disease is spreading, one measures the everyday new infection cases to infer the transmission rate in different regions. On a county-level modeling, one treats $3,141$ different counties in the US separately, and the parameter to be inferred has a dimension of at least $3,141$ (Li et al., 2020).

In this work, we focus on Monte Carlo sampling of log-concave probability distributions on $\mathbb{R}^d$, meaning the probability density can be written as $p(x) \propto e^{-f(x)}$ where a $f(x)$ is a convex function. The goal is to generate (approximately) i.i.d. samples according to the target probability distribution with density $p(x)$. Several sampling frameworks have been proposed in the literature, including importance sampling and sequential Monte Carlo (Geweke, 1989; Neal, 2001; Del Moral et al., 2006); ensemble methods (Reich, 2011; Iglesias et al., 2013); Markov chain Monte Carlo (MCMC) (Roberts and Rosenthal, 2004), including Metropolis-Hasting based MCMC (MH-MCMC) (Metropolis et al., 1953; Hastings, 1970; Roberts and Tweedie, 1996); Gibbs samplers (Geman and Geman, 1984; Casella and George, 1992); and Hamiltonian Monte Carlo (Neal, 1993; Duane et al., 1987). Langevin Monte Carlo (LMC) (Rossky et al., 1978; Parisi, 1981; Roberts and Tweedie, 1996) is a popular MCMC method that has received intense attention in recent years due to progress in the non-asymptotic analysis of its convergence properties (Durmus and Moulines, 2017; Dalalyan, 2017; Dalalyan and Karagulyan, 2019; Durmus et al., 2019).

Denoting by $x^m$ the location of the sample at $m$-th iteration, LMC obtains the next location as follows:
$$x^{m+1} = x^m - \nabla f(x^m)h + \sqrt{2h}\xi_d^m , \tag{1}$$

where $h$ is the time stepsize, and $\xi_d^m$ is drawn i.i.d. from $\mathcal{N}(0, I_d)$, where $I_d$ denotes identity matrix of size $d \times d$. LMC can be viewed as the Euler-Maruyama discretization of the following stochastic differential equation (SDE):

$$dX_t = -\nabla f(X_t)\, dt + \sqrt{2}\, dB_{t,d}\,, \tag{2}$$

where $B_{t,d}$ is a $d$-dimensional Brownian motion. It is well known that under suitable conditions, the distribution of $X_t$ converges exponentially fast to the target distribution (see e.g., (Markowich and Villani, 1999)). Since (1) approximates the SDE (2) with an $\mathcal{O}(h)$ discretization error, the probability distribution of $x^m$ produced by LMC (1) converges exponentially to the target distribution up to a discretization error (Dalalyan and Karagulyan, 2019).

A significant drawback of LMC is that the algorithm requires the evaluation of the full gradient at each iteration. This could be potentially very expensive in most practical problems. Indeed, when the analytical expression of the gradient is not available, each partial derivative component in the gradient needs to be computed separately, either through finite differencing or automatic differentiation (Baydin et al., 2017), so that the total number of such evaluations can be as many as $d$ times the number of required iterations. In the weather prediction and epidemiology problems discussed above, $f$ stands for the map from the parameter space of measured quantities via the underlying partial differential equations (PDEs), and each dimensional partial derivative calls for one forward and one adjoint PDE solve. Thus, $2d$ PDE solves are required in general at each iteration. Another example comes from the study of directed graphs with multiple nodes. Denote the nodes by $\mathcal{N} = \{1, 2, \ldots, d\}$ and directed edges by $\mathcal{E} \subset \{(i, j) : i, j \in \mathcal{N}\}$, and suppose there is a scalar variable $x_i$ associated with each node. When the function $f$ has the form $f(x) = \sum_{(i,j) \in \mathcal{E}} f_{ij}(x_i, x_j)$, the partial derivative of $f$ with respect to $x_i$ is given by

$$\frac{\partial f}{\partial x_i} = \sum_{j:(i,j) \in \mathcal{E}} \frac{\partial f_{ij}}{\partial x_i}(x_i, x_j) + \sum_{l:(l,i) \in \mathcal{E}} \frac{\partial f_{li}}{\partial x_i}(x_l, x_i)\,.$$

Note that the number of terms in the summations equals the number of edges that touch node $i$, the expected value of which is about $2/d$ times the total number of edges in the graph. Meanwhile, evaluation of the full gradient would require evaluation of both partial derivatives of each $f_{ij}$ for *all* edges in the graph. Hence, the cost difference between these two operations is a factor of order $d$.

In this paper, we study how to modify the updating strategies of LMC to reduce the numerical cost, with the focus on reducing dependence on $d$. In particular, we will develop and analyze a method called Random Coordinate Langevin Monte Carlo (RC-LMC). This idea is inspired by the random coordinate descent (RCD) algorithm from optimization (Nesterov, 2012; Wright, 2015). RCD is a version of Gradient Descent (GD) in which one coordinate (or a block of coordinates) is selected at random for updating along its negative gradient direction. In optimization, RCD can be significantly cheaper than GD, especially when the objective function is skewed and the dimensionality of the problem is high. In RC-LMC, we use the same basic strategy: At iteration $m$, a single coordinate of $x^m$ is randomly selected for updating, while all others are left unchanged.

Although each iteration of RC-LMC is cheaper than conventional LMC, more iterations are required to achieve the target accuracy, and delicate analysis is required to obtain bounds on the total cost. Analogous to optimization, the savings of RC-LMC by comparison with LMC depend on the structure of the dimensional Lipschitz constants. Under the assumption that there is a factor-of-$d$ difference in per-iteration costs, we compare our results with current results for classical LMC (Dalalyan and Karagulyan, 2019; Durmus et al., 2019) and conclude the following:

1. (Theorem 4.1) When the gradient of $f$ is Lipschitz but the Hessian is not, RC-LMC costs $\widetilde{O}(d^2/\epsilon^2)$ to get an $\epsilon$-accurate solution. Therefore, RC-LMC outperforms LMC, in terms of the computational cost, if $f$ is skewed and the dimension of the problem is high, as discussed in Remark 4.1. The optimal numerical cost in this setting is achieved when the probability of choosing the $i$-th direction is proportional to the $i$-th directional Lipschitz constant.

2. (Theorem 4.2) When both the gradient and the Hessian of $f$ are Lipschitz, RC-LMC requires $\widetilde{O}(d^{3/2}/\epsilon)$ iterations to achieve $\epsilon$ accuracy. On the other hand, the currently available result indicates that the classical LMC costs $\widetilde{O}(d^2/\epsilon)$. Thus, RC-LMC saves a factor of at least $d^{1/2}$ regardless of the stiffness structure of $f$, as discussed in Remark 4.2.

3. (Proposition 4.2) The $\widetilde{O}(d^{3/2}/\epsilon)$ complexity bound for RC-LMC is sharp when both the gradient and the Hessian of $f$ are Lipschitz.

(The notation $\widetilde{O}(\cdot)$ omits possible log terms.) We make three additional remarks. (a) Throughout the paper we assume that one element of the gradient is available at an expected cost of approximately $1/d$ of the cost of the full gradient evaluation. Although this property is intuitive, and often holds in many situations (such as the graph-based example presented above), it does not hold for all problems (Wright, 2015). (b) Besides replacing gradient evaluation by coordinate algorithms, one might also improve the dimension dependence of LMC by utilizing a more rapidly convergent method for the underlying SDEs than (2). One such possibility is to use underdamped Langevin dynamics, see e.g., (Rossky et al., 1978; Dalalyan and Riou-Durand, 2018; Cheng et al., 2018; Eberle et al., 2019; Shen and Lee, 2019; Cao et al., 2019), which can also be combined with coordinate sampling. For the clarity of presentation, we will focus only on LMC in this work and leave the extension to underdampped samplers to a future work. (c) It is also possible to reduce the cost of full gradient evaluation using stochastic gradient (Welling and Teh, 2011) or MALA-in-Gibbs sampling (Tong et al., 2020). However, both methods require specific forms of the objective function that are not considered in our work.

The paper is organized as follows. We present the RC-LMC algorithm in Section 2. Notations and assumptions on $f$ are listed in Section 3, where we also recall theoretical results for the classical LMC method. We present our main results regarding the numerical cost in Section 4 and numerical experiments in Section 5. Proofs of the main results are deferred to the Appendix.

## 2 RANDOM COORDINATE LANGEVIN MONTE CARLO

We introduce the Random Coordinate Langevin Monte Carlo (RC-LMC) method in this section. At each iteration, one coordinate is chosen at random and updated, while the other components of $x$ are unchanged. Specifically, denoting by $r^m$ the index of the random coordinate chosen at $m$-th iteration, we obtain $x_{r^m}^{m+1}$ according to a single-coordinate version of (1) and set $x_i^{m+1} = x_i^m$ for $i \neq r^m$.

The coordinate index $r^m$ can be chosen uniformly from $\{1, 2, \ldots, d\}$; but we will consider more general possibilities. Let $\phi_i$ be the probability of component $i$ being chosen, we denote the distribution from which $r^m$ is drawn by $\Phi$, where

$$\Phi := \{\phi_1, \phi_2, \ldots, \phi_d\}, \quad \text{where } \phi_i > 0 \text{ for all } i \text{ and } \sum_{i=1}^{d} \phi_i = 1. \tag{3}$$

The stepsize may depend on the choice of coordinate; we denote the stepsizes by $\{h_1, h_2, \ldots, h_d\}$ and assume that they do not change across iterations. In this paper, we choose $h_i$ to be inversely dependent on probabilities $\phi_i$, as follows:

$$h_i = \frac{h}{\phi_i}, \quad i = 1, 2, \ldots, d, \tag{4}$$

where $h > 0$ is a parameter that can be viewed as the expected stepsize. In Section 4.2-4.3, we will find the optimal form of $\Phi$ under different scenarios. The initial iterate $x^0$ is drawn from a distribution $q_0$, which can be any distribution that is easy to draw from (the normal distribution, for example). We present the complete method in Algorithm 1.

When we compare (5) with the classical LMC (1), we see that only one random coordinate is updated per iteration, meaning:

$$\nabla f(x^m) \to \partial_{r^m} f(x^m) e_{r^m}, \quad \xi_d^m \to \xi^m e_{r^m}$$

where $e_i$ is the unit vector for $i$-th direction and $\xi^m$ is drawn from $\mathcal{N}(0, 1)$. Define the elapsed time at $m$-th iteration as

$$T^m := \sum_{n=0}^{m-1} h_{r^n}, \quad \text{and} \quad T^0 := 0, \tag{6}$$

then for $t \in (T^m, T^{m+1}]$, the updating formula (5) can be viewed as the Euler approximation to the following SDE:

$$\begin{cases} X_{r^m}(t) = X_{r^m}(T^m) - \displaystyle\int_{T^m}^{t} \partial_{r^m} f(X(s)) \, ds + \sqrt{2} \int_{T^m}^{t} \, dB_s, \\ X_i(t) = X_i(T^m), \quad \forall i \neq r^m, \end{cases} \tag{7}$$

---

**Algorithm 1 Random Coordinate Langevin Monte Carlo** (RC-LMC)

---

**Input:** Coordinate distribution $\Phi := \{\phi_1, \phi_2, \ldots, \phi_d\}$; parameter $h > 0$ and stepsize set $\{h_1, h_2, \ldots, h_d\}$ defined in (3)–(4); $M$ (stop index).

Sample $x^0$ from an initial distribution $q_0$
**for** $m = 0, 1, 2, \ldots M - 1$ **do**
    1. Draw $r^m \in \{1, \ldots, d\}$ according to probability distribution $\Phi$;
    2. Draw $\xi^m$ from $\mathcal{N}(0, 1)$;
    3. Update $x^{m+1}$ by

$$x_i^{m+1} = \begin{cases} x_i^m - h_i \partial_i f(x^m) + \sqrt{2h_i}\, \xi^m, & i = r^m \\ x_i^m, & i \neq r^m. \end{cases} \tag{5}$$

**end for**
**return** $x^M$

---

where $B_t$ is a 1-dimensional Brownian motion. We will show in Section 4.1 that the SDE preserves the invariant measure, that is, $X(t) \sim p$ for any $t > 0$ if $X(0) \sim p$, and it is ergodic. The invariant measure of RC-LMC, which can be viewed as a discretized version of the SDE, is not exactly $p$, due to the unavoidable discretization error.

## 3 NOTATIONS, ASSUMPTIONS AND CLASSICAL RESULTS

We unify notations and assumptions in this section, and summarize and discuss the classical results on LMC. Throughout the paper, to quantify the distance between two probability distributions, we use the Wasserstein distance defined by

$$W(\mu, \nu) = \left( \inf_{(X,Y) \in \Gamma(\mu,\nu)} \mathbb{E}|X - Y|^2 \right)^{1/2},$$

where $\Gamma(\mu, \nu)$ is the set of distribution of $(X, Y) \in \mathbb{R}^{2d}$ whose marginal distributions, for $X$ and $Y$ respectively, are $\mu$ and $\nu$. The distributions in $\Gamma(\mu, \nu)$ are called the *couplings* of $\mu$ and $\nu$. Due to the use of power 2 in the definition, this is sometimes called the Wasserstein-2 distance. Here and in the sequel, we use $|\cdot|$ to denote the Euclidean norm of a vector.

We assume that $f$ is strongly convex, so that $p$ is strongly log-concave. We obtain results under two different assumptions: First, Lipschitz continuity of the gradient of $f$ (Assumption 3.1) and second, Lipschitz continuity of the Hessian of $f$ (Assumption 3.2 together with Assumption 3.1).

**Assumption 3.1.** *The function $f$ is twice differentiable, $f$ is $\mu$-strongly convex for some $\mu > 0$ and its gradient $\nabla f$ is $L$-Lipschitz. That is, for all $x, x' \in \mathbb{R}^d$, we have*

$$f(x) - f(x') - \nabla f(x')^\top (x - x') \geq \frac{\mu}{2}|x - x'|^2, \tag{8}$$

*and*

$$|\nabla f(x) - \nabla f(x')| \leq L|x - x'|. \tag{9}$$

It is an elementary consequence of (8) that

$$(\nabla f(x') - \nabla f(x))^\top (x' - x) \geq \mu|x' - x|^2, \quad \text{for all } x, x' \in \mathbb{R}^d. \tag{10}$$

Since each coordinate direction plays a distinct role in RC-LMC, we distinguish the Lipschitz constants in each such direction. When Assumption 3.1 holds, partial derivatives in all coordinate directions are also Lipschitz. Denoting them as $L_i$ for each $i = 1, 2, \ldots, d$, we have

$$|\partial_i f(x + t e_i) - \partial_i f(x)| \leq L_i|t| \tag{11}$$

for any $x \in \mathbb{R}^d$ and any $t \in \mathbb{R}$. We further denote $L_{\max} := \max_i L_i$ and define condition numbers as follows:

$$\kappa = L/\mu \geq 1, \quad \kappa_i = L_i/\mu \geq 1, \quad \kappa_{\max} = \max_i \kappa_i. \tag{12}$$

As shown in (Wright, 2015), we have

$$L_i \leq L_{\max} \leq L \leq dL_{\max}, \quad \kappa_i \leq \kappa_{\max} \leq \kappa \leq d\kappa_{\max}. \tag{13}$$

These assumptions together imply that the spectrum of the Hessian is bounded above and below for all $x$, specifically, $\mu I_d \preceq \nabla^2 f(x) \preceq LI_d$ and $[\nabla^2 f(x)]_{ii} \leq L_i \leq L_{\max}$ for all $x \in \mathbb{R}^d$.

Both upper and lower bounds of $L$ in term of $L_{\max}$ in (13) are tight. If $\nabla^2 f$ is a diagonal matrix, then $L_{\max} = L$, both being the biggest eigenvalue of $\nabla^2 f$. Thus, $\kappa_{\max} = \kappa$ in this case. This is the case in which all coordinates are independent of each other, for example $f = \sum_i \lambda_i x_i^2$. On the other hand, if $\nabla^2 f = \mathbf{e} \cdot \mathbf{e}^\top$ where $\mathbf{e} \in \mathbb{R}^d$ satisfies $\mathbf{e}_i = 1$ for all $i$, then $L = dL_{\max}$ and $\kappa = d\kappa_{\max}$. This is a situation in which $f$ is highly skewed, that is, $f = (\sum_i x_i)^2/2$.

The next assumption concerns higher regularity for $f$.

**Assumption 3.2.** *The function $f$ is three times differentiable and $\nabla^2 f$ is H-Lipschitz, that is*

$$\|\nabla^2 f(x) - \nabla^2 f(x')\|_2 \leq H|x - x'|, \quad \text{for all } x, x' \in \mathbb{R}^d. \tag{14}$$

When this assumption holds, we further define $H_i$ to satisfy

$$|\partial_{ii} f(x + t\mathbf{e}_i) - \partial_{ii} f(x)| \leq H_i|t|, \tag{15}$$

for any $i = 1, 2, \ldots, d$, all $x \in \mathbb{R}^d$, and all $t \in \mathbb{R}$, where $\partial_{ii} f$ is $[\nabla^2 f(x)]_{ii}$, the $(i, i)$ diagonal entry of the Hessian matrix $\nabla^2 f$.

We summarize existing results for the classical LMC in the following theorem.

**Theorem 3.1** ((Durmus et al., 2019, Theorem 9), (Dalalyan and Karagulyan, 2019, Theorem 5)). *Let $q_m$ be the probability distribution of the $m$-th iteration of LMC (1), and $p$ be the target distribution. Using the notation $W_m := W(q_m, p)$, we have the following:*

- *Under Assumption 3.1, let $h \leq 1/L$, we have*

$$W_m \leq \exp\left(-\mu h m/2\right) W_0 + 2(\kappa h d)^{1/2}; \tag{16}$$

- *Under Assumptions 3.1 and 3.2, let $h < 2/(\mu + L)$, we have*

$$W_m \leq \exp\left(-\mu h m\right) W_0 + \frac{Hhd}{2\mu} + 3\kappa^{3/2}\mu^{1/2}hd^{1/2}. \tag{17}$$

This theorem yields stopping criteria for the number of iterations $M$ to achieve a user-defined accuracy of $\epsilon$. When the gradient of $f$ is Lipschitz, to achieve $\epsilon$-accuracy, we can require both terms on the right hand side of (16) to be smaller than $\epsilon/2$, which occurs when

$$h = \Theta(\epsilon^2/d\kappa), \quad M = \Theta\left(\frac{1}{\mu h}\log\left(\frac{W_0}{\epsilon}\right)\right) = \Theta\left(\frac{d\kappa}{\mu\epsilon^2}\log\left(\frac{W_0}{\epsilon}\right)\right), \tag{18}$$

leading to a cost of $\widetilde{O}(d^2\kappa/(\mu\epsilon^2))$ evaluations of gradient components (when we assume that each full gradient can be obtained at the cost of $d$ individual components of the gradient). When both the gradient and the Hessian are Lipschitz, to achieve $\epsilon$-accuracy, we require all three terms on the right hand side of (17) to be smaller than $\epsilon/3$. Assuming $d \gg 1$ and all other constants are $O(1)$, we thus obtain

$$h = \Theta(\epsilon\mu/(dH + d^{1/2}L^{3/2})), \quad M = \Theta\left(\frac{dH + d^{1/2}L^{3/2}}{\mu^2\epsilon}\log\left(\frac{W_0}{\epsilon}\right)\right), \tag{19}$$

which yields a cost of $\widetilde{O}(d^2 H/(\mu^2\epsilon))$ evaluations of gradient components. Here $A = \Theta(B)$ denotes $cB \leq A \leq CB$ for some absolute constant $c$ and $C$.

## 4 MAIN RESULTS

We discuss the main results from two perspectives. In Section 4.1 we examine the convergence of the underlying SDE (7), laying the foundation for the convergence in the discrete setting. We then build upon this result and show the convergence of the RC-LMC algorithm in Section 4.2 and 4.3 under two different assumptions. We show in Section 4.4 that when both Assumption 3.1 and 3.2 are satisfied, our bound is tight with respect to $d$ and $\epsilon$.

### 4.1 CONVERGENCE OF THE SDE (7)

To study the convergence of (7), we first let $X^m = X(T^m)$ and denote the probability filtration by $\mathcal{F}^m = \left\{ x^0, r^{n \leq m}, B_{s \leq T^m} \right\}$. Then $\{X^m\}_{m=0}^{\infty}$ is a Markov chain and the following proposition shows its geometric ergodicity.

**Proposition 4.1.** *Let $X^m = X(T^m)$ solve* (7). *If $f$ satisfies Assumption 3.1 and $h \leq \frac{\mu \min\{\phi_i\}}{4+8L^2+32L^4}$, then $p(x)$ is the stationary probability density of the Markov chain $\{X^m\}_{m=0}^{\infty}$.*

See proof in Appendix A. Under some mild conditions, we can further prove that the solution to the SDE converges to the target distribution exponentially (Proposition A.1). Since $x^m$, the samples generated by the algorithm can be viewed as discrete version of $X^m$, the algorithm then is expected to converge up to a discretization error as well. This is indeed shown in the upcoming two subsections, where we document the non-asymptotic convergence rate, and calculate the complexity of the algorithm.

### 4.2 CONVERGENCE OF RC-LMC. CASE 1: LIPSCHITZ GRADIENT

Under Assumption 3.1, we have the following result. The proof can be found in Appendix B.

**Theorem 4.1.** *Assume $f$ satisfies Assumption 3.1, and $h_i = h/\phi_i$ with $h \leq \frac{\mu \min\{\phi_i\}}{8L^2}$. Let $q_m$ be the probability distribution of $x^m$ computed in* (5), *let $p$ be the target distribution, and denote $W_m := W(q_m, p)$. Then we have*

$$W_m \leq \exp\left(-\frac{\mu h m}{4}\right) W_0 + \frac{5 h^{1/2}}{\mu} \sqrt{\sum_{i=1}^{d} \frac{L_i^2}{\phi_i}} . \tag{20}$$

We make a few comments here: (1) the requirement on $h$ is rather weak. When both $\mu$ and $L$ are moderate (both $O(1)$ constants), the requirement is essentially $h \lesssim 1/d$. (2) The estimate (20) consists of two terms. The first is an exponentially decaying term and the second comes from the variance of random coordinate selection. If we assume all Lipschitz constants $L_i$ are of $O(1)$, this remainder term is roughly $O(h^{1/2}d)$. (3) The theorem suggests a stopping criterion: to have $W_M \leq \epsilon$, we roughly need $h < \epsilon^2/d^2$, and $M = \widetilde{O}(d^2/\epsilon^2)$, assuming $L_i = O(1)$. In terms of $\epsilon$ and $d$ dependence, this puts $M$ at the same order as (18), as required by the classical LMC.

Theorem 4.1 holds for all choices of $\{\phi_i\}$ satisfying (3). From the explicit formula (20) we can choose $\{\phi_i\}$ to minimize the right-hand side of the bound. Nesterov (2012) proposed distributions $\Phi$ that depend on the dimensional Lipschitz constants $L_i$, $i = 1, 2, \ldots, d$ from (11). For $\alpha \in \mathbb{R}$, we can let $\phi_i(\alpha) \propto L_i^\alpha$, specifically,

$$\phi_i(\alpha) := \frac{L_i^\alpha}{\sum_j L_j^\alpha} , \quad \text{and} \quad \Phi(\alpha) := \{\phi_1(\alpha), \phi_2(\alpha), \ldots, \phi_d(\alpha)\} . \tag{21}$$

Note that when $\alpha = 0$, $\phi_i(0) = 1/d$ for all $i$: the uniform distribution among all coordinates. When $\alpha > 0$, the directions that with larger Lipschitz constants have higher probability to be chosen. Since $h_i = h/\phi_i$, one uses smaller stepsizes for stiffer directions. (On the other hand, when $\alpha < 0$, the directions with larger Lipschitz constants are less likely to be chosen, and the stepsizes are larger in stiffer directions, a situation that is not favorable and should be avoided.) The following corollary discusses various choices of $\alpha$ and the corresponding computational cost.

**Corollary 4.1.** *Under the same conditions as in Theorem 4.1, with $\phi_i = \phi_i(\alpha)$ defined in* (21), *the number of iterations $M$ required to attain $W_M \leq \epsilon$ is $M = \Theta\left(\frac{K_{2-\alpha} K_\alpha}{\mu \epsilon^2} \log\left(\frac{W_0}{\epsilon}\right)\right)$, where $K_\alpha = \sum_{i=1}^{d} \kappa_i^\alpha$. This cost is optimized when $\alpha = 1$, for which we have*

$$M = \Theta\left(\frac{(\sum_i \kappa_i)^2}{\mu \epsilon^2} \log\left(\frac{W_0}{\epsilon}\right)\right) . \tag{22}$$

See proof in Appendix B. We note that the initial error $W_0$ enters through a $\log$ term and is essentially negligible.

**Remark 4.1.** *We now compare the numerical cost of RC-LMC and LMC in Case 1. We separate the discussion on uniform sampling ($\phi_i = 1/d$) and the optimal sampling ($\phi_i \propto L_i$) below.*

*– Optimal sampling: According to Corollary 4.1, the optimal sampling strategy is achieved when $\alpha = 1$, meaning $\phi_i \propto L_i$. In this case, we compare (22) with (18), adjusting (18) by a factor of $d$ to account for the higher cost per iteration. RC-LMC has more favorable computational cost if*

$$d^2 \kappa \geq \left( \sum_i \kappa_i \right)^2 .$$

*Considering $\kappa_i \leq \kappa_{\max} \leq \kappa \leq d\kappa_{\max}$, as presented in (13), this is guaranteed if $\kappa \geq \kappa_{\max}^2$. In the regime when $\kappa \sim d\kappa_{\max}$ this holds so long as $d > \kappa_{\max}$, meaning the dimension of the problem is high. And in the regime when $\kappa_{\max} \sim \kappa$, RC-LMC still outperforms when $\kappa_i$ decreases fast. One example is to set $f(x) = dx_1^2 + \sum_{i=2}^{d} x_i^2$ with $d \gg 1$.*

*– Uniform sampling: Uniform sampling means $\phi_i = 1/d$ for all $i$, with $\alpha = 0$ in Corollary 4.1. This leads to a cost of $\Theta\left( \frac{\sum \kappa_i^2}{\mu \epsilon^2} \log\left( \frac{W_0}{\epsilon} \right) \right)$. Comparing with (18) adjusted by a factor of $d$, we see that RC-LMC still has a more favorable computational cost if*

$$d^2 \kappa \geq \sum_i \kappa_i^2 .$$

*As in the optimal case, this happens when $f$ is highly skewed.*

Our proof of Theorem 4.1 follows from a coupling approach similar to that used by Dalalyan and Karagulyan (2019) for LMC. We emphasize that for the coordinate algorithm, we need to overcome the additional difficulty that the process of each coordinate is not contracting on the SDE (7) level. This is a different situation from the classical LMC (Dalalyan and Karagulyan, 2019) whose corresponding SDE (2) already provides the contraction property and thus only the discretization error needs to be considered. Despite this, the algorithm RC-LMC still enjoys the contraction property that ensures that the distance between two different trajectories following the algorithm contract. However, this contraction property is not component-wise, so we need to choose Young's constant wisely and take summation of every coordinate. The summation will also produce some extra terms, which we need to bound. Dalalyan and Karagulyan (2019) obtains an estimate for the cost of the classical LMC of $\widetilde{O}(d^2\kappa^2/(\mu\epsilon^2))$. Compared with this estimate, our estimate for the cost of RC-LMC is always cheaper (since $\kappa^2 \geq \kappa_{\max}^2$). The improved estimate of the cost of LMC (18) was obtained by Durmus et al. (2019) using a quite different approach based on optimal transportation. It is not clear whether their technique can be adapted to the coordinate setting to obtain an improved estimate.

### 4.3 Convergence of RC-LMC. Case 2: Lipschitz Hessian

We now assume that Assumption 3.1 and 3.2 hold, that is, both the gradient and the Hessian of $f$ are Lipschitz continuous. In this setting, we obtain the following improved convergence estimate. The proof can be found in Appendix C.

**Theorem 4.2.** *Assume $f$ satisfies Assumptions 3.1 and 3.2 and let $h_i = h/\phi_i$, with $h \leq \frac{\mu \min\{\phi_i\}}{8L^2}$. Denoting by $q_m(x)$ the probability density function of $x^m$ computed from (5) and by $p$ the target distribution, and letting $W_m := W(q_m, p)$, we have:*

$$W_m \leq \exp\left( -\frac{\mu h m}{4} \right) W_0 + \frac{3h}{\mu} \sqrt{\sum_{i=1}^{d} \frac{(L_i^3 + H_i^2)}{\phi_i^2}} . \tag{23}$$

We see again two terms in the bound, an exponentially decaying term and a variance term. Assuming all Lipschitz constants are $O(1)$, the variance term is of $O(hd^{3/2})$. By comparing with Theorem 4.1, we see that $\epsilon$ error can be achieved with the looser stepsize requirement $h \lesssim \frac{\epsilon}{d^{3/2}}$.

By choosing $\{\phi_i\}$ to optimize the bound in Theorem 4.2, we obtain the following corollary.

**Corollary 4.2.** *Under the same conditions as in Theorem 4.2, the optimal choice of $\{\phi_i\}$ is to set:*

$$\phi_i = \frac{\left(L_i^3 + H_i^2\right)^{1/3}}{\sum_{i=1}^d \left(L_i^3 + H_i^2\right)^{1/3}}\,.$$

*For this choice, the number of iterations $M$ required to guarantee $W_M \leq \epsilon$ satisfies*

$$M = \Theta\left(\frac{\left(\sum_{i=1}^d \left(L_i^3 + H_i^2\right)^{1/3}\right)\left(\sum_{i=1}^d \left(L_i^3 + H_i^2\right)^{1/3}\right)^{1/2}}{\mu^2 \epsilon} \log\left(\frac{W_0}{\epsilon}\right)\right). \quad (24)$$

*If $\mu$, $\kappa_i$ and $H_i$ are all constants of $O(1)$, then the total cost is $\widetilde{O}(d^{3/2}/\epsilon)$ regardless of the choice of $\{\phi_i\}$.*

**Remark 4.2.** *We now compare RC-LMC with LMC in Case 2 using Theorem 4.2 and Corollary 4.2. We still separate the discussion on optimal sampling and uniform sampling strategy.*

*– Optimal sampling: This is to set $\phi_i$ as stated in Corollary 4.2. Comparing the cost shown in (24) and the cost of LMC ( (19) adjusted by a factor of $d$ to account for the higher cost per iteration), we see that RC-LMC always has a more favorable computational cost since*

$$d^2 H + d^{3/2} L^{3/2} \geq d^{3/2}(L^3 + H^2)^{1/2}\,.$$

*(Here we relaxed (24) using $L_i \leq L, H_i \leq H$.)*

*– Uniform sampling: This is to set $\phi_i = 1/d$ in (23). Then the total cost of RC-LMC is*

$$M = \Theta\left(\frac{\left(\sum_{i=1}^d \left(L_i^3 + H_i^2\right)^{1/3}\right)^{3/2}}{\mu^2 \epsilon} \log\left(\frac{W_0}{\epsilon}\right)\right),$$

*according to Corollary4.2. Comparing with (19) adjusted by a factor of $d$ as the cost for LMC, and use the fact that $L_i \leq L, H_i \leq H$, it is clear that RC-LMC is always cheaper, similar to the optimal case.*

*Suppose $L$ and $H$ are all constants of $O(1)$, then the cost of RC-LMC is roughly $\widetilde{O}(d^{3/2}/\epsilon)$, while the classical LMC requires $\widetilde{O}(d^2/\epsilon)$, according to Dalalyan and Riou-Durand (2018). This represents a savings factor of $d^{1/2}$, regardless of the structure of $f$.*

### 4.4 TIGHTNESS OF THE COMPLEXITY BOUND

When both the gradient and the Hessian are Lipschitz, we claim that estimate $\widetilde{O}(d^{3/2}/\epsilon)$ obtained in Corollary 4.2 is tight. An example is presented in the following proposition.

**Proposition 4.2.** *Let $\phi_i = 1/d$ for all $i$, and set the initial distribution and the target distribution to be:*

$$q_0(x) = \frac{1}{(4\pi)^{d/2}} \exp(-|x - \mathsf{e}|^2/4)\,, \quad p(x) = \frac{1}{(2\pi)^{d/2}} \exp(-|x|^2/2)\,, \quad (25)$$

*where $\mathsf{e} \in \mathbb{R}^d$ satisfies $\mathsf{e}_i = 1$ for all $i$. Let $q_m$ be the probability distribution of $x^m$ generated by Algorithm 1, and denote $W_m := W(q_m, p)$. Then we have*

$$W_m \geq \exp\left(-2mh\right)\frac{\sqrt{d}}{3} + \frac{d^{3/2}h}{6}\,, \quad m \geq 1\,. \quad (26)$$

*In particular, to have $W_M \leq \epsilon$, one needs at least $M = \widetilde{O}(d^{3/2}/\epsilon)$.*

See proof in Appendix D.

## 5 NUMERICAL RESULTS

We provide some numerical results in this section. Since it is extremely challenging to estimate the Wasserstein distance between two distributions in high dimensions, we demonstrate instead the convergence of estimated expectation for a given observable. Denoting by $\{x^{(i),M}\}_{i=1}^N$ the list of $N$ samples, with each of them computed through Algorithm 1 independently with $M$ iterations, we define the error as follows:

$$\text{Error}_{M,N} = \left\| \frac{1}{N} \sum_{i=1}^N \psi(x^{(i),M}) - \mathbb{E}_{p\mathsf{x}}(\psi) \right\|_2, \tag{27}$$

where $\psi$ is a matrix function and $\mathbb{E}_p(\psi)$ is the expectation of $\psi$ under the target distribution $p$. As $h \to 0$ and $Mh \to \infty$, we have $W_M \to 0$, and $x^{(i),M}$ can be regarded as approximately sampled from $p$. According to the central limit theorem, we have $\lim_{h\to 0, Mh\to\infty} \text{Error}_{M,N} = O(1/\sqrt{N})$.

In this example, we set the target and initial distributions to be Gaussian $p(x) \propto p_1(\mathsf{x})p_2(x)$ and $q_0(x) \propto p_1(\mathsf{x} - \mathsf{e})p_2(x)$ with

$$p_1(\mathsf{x}) = \exp\left( -\frac{1}{2}\mathsf{x}\left(\mathsf{T} + (d/10)I\right)^\top \left(\mathsf{T} + (d/10)I\right)\mathsf{x}^\top \right), \quad p_2 = \exp\left( -\frac{1}{2}\sum_{i=11}^{100} |x_i|^2 \right),$$

where $\mathsf{x} = (x_1, x_2, \ldots, x_{10})^\top$, $\mathsf{e} = (1, 1, \ldots, 1)^\top \in \mathbb{R}^{10}$, $I$ is the identity matrix and $\mathsf{T}$ is a random matrix with each entry i.i.d. drawn from $\mathcal{N}(0, 1)$. We run the simulation with $N = 10^5$, and we compute $\text{Error}_M$ with $\psi(x) = \mathsf{xx}^\top$. This measures the spectral norm of the covariance matrix of the first 10 entries.

The results are shown in Figure 1. We run RC-LMC with time stepsize $h = 10^{-5}$ and $\alpha = 1$, following (21). It is unclear what stepsize $h$ to choose for LMC to yield a fair comparison. Bearing in mind that $d = 100$ in this example, so that the per-iteration cost of LMC is 100 times of that of RC-LMC, we try first $h = 10^{-3}$. It is clear that RC-LMC, presented by the purple dashed line, achieves lower error than LMC at the same cost, before achieving the error plateau. Next, we try smaller choices of $h$ in LMC. The choices $h = .0008$ and $h = .0005$ yield slower decay rates (see the red (star) and yellow (circle) lines, respectively), but lower error plateaus as well, meaning that the saturation error is smaller. However, computation required to reach these plateaus is longer, and the plateaus are still higher than for RC-LMC.

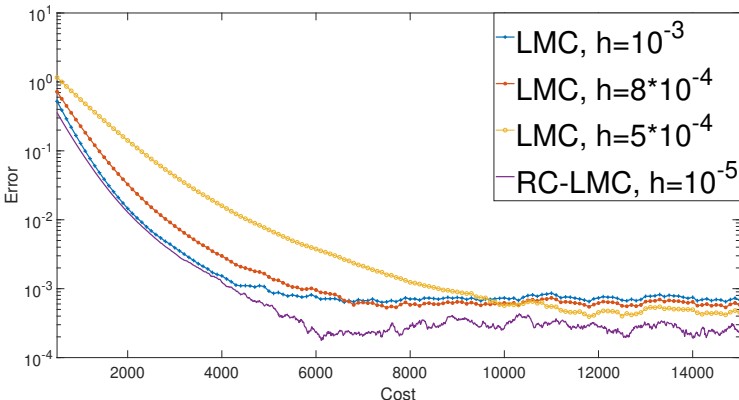

Figure 1: The decay of error with respect to the cost (number of $\partial f$ calculations).

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

# A   PROOF OF PROPOSITION 4.1

We recall the SDE (7):

$$
\begin{cases}
X_{r^m}(t) = X_{r^m}(T^m) - \displaystyle\int_{T^m}^{t} \partial_{r^m} f(X(s))\,\mathrm{d}s + \sqrt{2}\int_{T^m}^{t} \mathrm{d}B_s\,, \\
X_i(t) = X_i(T^m)\,, \quad \forall i \neq r^m\,,
\end{cases}
\tag{28}
$$

where $r^m$ is randomly selected from $1, \ldots, d$. Moreover, recall that $X^{m+1} = X\left(T^{m+1}\right)$ is a Markovian process. We denote its transition kernel by $\Xi$, meaning that

$$
X^{m+1} \overset{d}{=} \Xi(X^m, \cdot)\,.
$$

Moreover, we denote $\Xi^n$ the $n$-step transition kernel. Proposition 4.1 is a consequence of the following Proposition.

**Proposition A.1.** *Denote $\Pi^m$ the probability distribution of $X^m$ and $\Pi$ be the probability distribution induced by $p(x)$, then under the conditions of Proposition 4.1, we have*

- *$\Pi$ is the stationary distribution of the Markov chain $\{X^m\}_{m=0}^{\infty}$.*

- *If the second moment of $\Pi^0$ is finite and $X^0$ is drawn from $\Pi^0$, then there are constants $R > 0$ and $r > 1$, independent of $m$, such that for any $m \geq 0$ we have*

$$
d_{TV}\left(\Pi^m, \Pi\right)\,\mathrm{d}x \leq R r^{-m}\,.
\tag{29}
$$

**Remark A.1.** *According to Mattingly et al. (2002), the constants $R$ and $r$ do not depend on $m$, but their dependence on other parameters such as $h$, $d$, and $L$ is hard to trace. This contrasts with the results in Dalalyan and Karagulyan (2019) for the classical Langevin dynamics, which are built upon the contraction property. The new complication comes mainly from the complicated coordinate selection process, making the contraction property no longer available. Nor can we claim sharpness of the theorem. In fact, unlike in Dalalyan and Karagulyan (2019); Xu et al. (2018), where the authors directly studied LMC, we discuss here only convergence of the SDE, the continuous version of RC-LMC. The explicit dependencies of the convergence rate here are unimportant, and we allow the results to be loose. Non-asymptotic convergence results of the algorithms are presented in Section 4.2 and 4.3.*

To prove Proposition A.1, we need to introduce the following lemma:

**Lemma A.1.** *Under conditions of Theorem 4.1, there are constants $R_1 > 0, r_1 > 1$, such that for any $z^0 \in \mathbb{R}^d$*

$$
\sup_{A \in \mathcal{B}(\mathbb{R}^d)} \left| \Xi^{md}(z^0, A) - \int_A p(x)\,\mathrm{d}x \right| \leq \left( |z^0 - x^*|^2 + 1 \right) R_1 r_1^{-m}\,,
\tag{30}
$$

*where $x^*$ is the minimal point of $f(x)$ and $\Xi$ is the transition kernel for $\{X^m\}_{m=0}^{\infty}$.*

We postpone the proof of Lemma A.1 to Section A.1. Now, we are ready to prove the proposition.

*Proof of Proposition A.1 (Proposition 4.1).*   To prove the first bullet point of Proposition A.1, we assume the distribution of $X^m$ is $\Pi$ and we need to prove:

*For any choice of $r^m$, the conditional distribution of $X^{m+1}$ is also $\Pi$.*

Without loss of generality, we consider $r^m = 1$. Under this condition, we have the following.

- The distribution of $X_{2 \leq j \leq d}(t)$ between $[T^m, T^{m+1}]$ is preserved.

- For fixed $z_2, z_3, \ldots, z_d$, the stationary density of SDE

$$\mathrm{d}z = -\partial_1 f(z, z_2, z_3, \ldots, z_d) \, \mathrm{d}t + \sqrt{2} \, \mathrm{d}B_s \,, \tag{31}$$

is $\frac{\exp(-f(z, z_2, \ldots, z_d))}{\int \exp(-f(z, z_2, \ldots, z_d) \, \mathrm{d}z)}$. This implies that the conditional distribution of $X_1(t)$ with fixed $X_{2 \le j \le d}(t)$ is also preserved.

Combining these two points, we find that under condition $r^m = 1$, the conditional distribution of $X^{m+1}$ is $\Pi$, which further proves that $\Pi$ is the stationary distribution and Proposition 4.1 holds.

To show (29), we take the expectation of (30) using $\Pi^0$, then we can obtain that for any $A \in \mathcal{B}_{\mathbb{R}^d}$ hat

$$
\begin{aligned}
\left| \Pi^{md}(A) - \Pi(A) \right| &\overset{(I)}{=} \left| \mathbb{E}_{\Pi^0} \left( \Xi^{md}(\cdot, A) \right) - \mathbb{E}_{\Pi^0} \left( \int_A p(x) \, \mathrm{d}x \right) \right| \\
&\overset{(II)}{\le} \mathbb{E}_{\Pi^0} \left( \left| \Xi^{md}(z, A) - \int_A p(x) \, \mathrm{d}x \right| \right) \\
&\overset{(III)}{\le} R_1 r_1^{-m} \int_{\mathbb{R}^d} \left( |z - z^*|^2 + 1 \right) q_0(z) \, \mathrm{d}z < C_0 r_1^{-m} \,,
\end{aligned}
$$

where we use $X^{md} \overset{d}{=} \Xi^{md}(X^0, \cdot)$ and $\Pi(A) = \int_A p(x) \, \mathrm{d}x$ in $(I)$, $\Pi^0$ is a non-negative measure in $(II)$ and (30) in $(III)$. Since this is true for all $A \in \mathcal{B}_{\mathbb{R}^d}$, we have

$$d_{TV}(\Pi^{md}, \Pi) = \sup_{A \in \mathcal{B}_{\mathbb{R}^d}} \left| \Pi^{md}(A) - \Pi^m(A) \right| < C_0 r_1^{-m} \,. \tag{32}$$

By using (28) with Itô's formula, we have

$$
\begin{aligned}
\frac{\mathrm{d}\mathbb{E}|X_{r^m}(t)|^2}{\mathrm{d}t} &= -2\mathbb{E} \left( \partial_{r^m} f(X_{r^m}(t)) X_{r^m}(t) \right) + 2 \le 2 + \mathbb{E}|\partial_{r^m} f(X_{r^m}(t))|^2 + \mathbb{E}|X_{r^m}(t)|^2 \\
&\le 2 + L_{r^m}^2 \mathbb{E}|X_{r^m}(t) - x_{r^m}^*|^2 + \mathbb{E}|X_{r^m}(t)|^2 \le C_{1,r^m} \mathbb{E}|X_{r^m}(t)|^2 + C_{2,r^m} \,,
\end{aligned}
$$

where $C_{1,r^m}$ and $C_{2,r^m}$ are constants that depend only on $x^*$ and $L_{r^m}$. From Grönwall's inequality, we obtain

$$\mathbb{E} \left( |X_i^{m+1}|^2 \big| r^m = i \right) \le \exp(C_{1,i} h_i) \left[ \mathbb{E}(|X_i^m|^2) + C_{2,i} h_i \right] \,, \quad \text{for all } i = 1, 2, \ldots, d.$$

Then, if $\mathbb{E}|X^m|^2 < \infty$, we have for any $i = 1, 2, \ldots, d$ that

$$
\begin{aligned}
\mathbb{E} \left( |X_i^{m+1}|^2 \right) &= \frac{1}{d} \mathbb{E} \left( |X_i^{m+1}|^2 \big| r^m = i \right) + \left( 1 - \frac{1}{d} \right) \mathbb{E} \left( |X_i^{m+1}|^2 \big| r^m \ne i \right) \\
&\le \frac{1}{d} \exp(C_{1,i} h_i) \left[ \mathbb{E}(|X_i^m|^2) + C_{2,i} h_i \right] + \left( 1 - \frac{1}{d} \right) \mathbb{E}(|X_i^m|^2) < \infty \,,
\end{aligned}
$$

which implies $\mathbb{E}|X^{m+1}|^2 < \infty$. Therefore, if $\Pi^0$ has finite second moment, then $\Pi^i$ all have finite second moments for $i = 1, \ldots, d-1$. Suppose the initial data is drawn from $\Pi^i$ for $i < d$, then taking the expectation of (30) and using (32), we obtain

$$d_{TV}(\Pi^{md+i}, \Pi) \le C_i r_1^{-m} \,,$$

where $C_i$ is a constant. This bound holds true for all $0 \le i \le d-1$, we set $R = (\max_i C_i) r_1$ and $r = r_1^{1/d}$ to obtain (29). $\qquad \square$

### A.1 Proof of Lemma A.1

Before we prove the Lemma, we first recall a result from (Mattingly et al., 2002) for the convergence of Markov chain using Lyapunov condition together with minorization condition.

**Theorem A.1.** *[(Mattingly et al., 2002, Theorem 2.5)] Let $\{X^n\}_{n=0}^{\infty}$ denote the Markov chain on $\mathbb{R}^d$ with transition kernel $\Xi$ and filtration $\mathcal{F}^n$. Let $\{X^n\}_{n=0}^{\infty}$ satisfy the following two conditions:*

*Lyapunov condition: There is a function* $L : \mathbb{R}^d \to [1, \infty)$, *with* $\lim_{x \to \infty} L(x) = \infty$, *and real numbers* $\alpha \in (0, 1)$, *and* $\beta \in [0, \infty)$ *such that*

$$\mathbb{E}\left(L(X^{n+1}) \big| \mathcal{F}^n\right) \leq \alpha L(X^n) + \beta.$$

*Minorization condition: For L from the Lyqpunov condition, define the set* $C \subset \mathbb{R}^d$ *as follows:*

$$C = \left\{ x \in \mathbb{R}^d \mid L(x) \leq \frac{2\beta}{\gamma - \alpha} \right\}, \tag{33}$$

*for some* $\gamma \in (\alpha^{1/2}, 1)$. *Then there exists an* $\eta > 0$ *and a probability measure* $\mathcal{M}$ *supported on C (that is,* $\mathcal{M}(C) = 1$), *such that*

$$\Xi(x, A) \geq \eta \mathcal{M}(A), \quad \forall A \in \mathcal{B}(\mathbb{R}^d), \ x \in C.$$

*Under these conditions, the Markov chain* $\{X^n\}_{n=0}^{\infty}$ *has a unique invariant measure* $\pi$. *Furthermore, there are constants* $r \in (0, 1)$ *and* $R \in (0, \infty)$ *such that, for any* $z^0 \in \mathbb{R}^d$, *we have*

$$\sup_{A \in \mathcal{B}(\mathbb{R}^d)} \left| \Xi^n(z^0, A) - \pi(A) \right| \leq L(z^0) R r^{-n}. \tag{34}$$

To use this result to prove Lemma A.1, we will consider the $d$-step chain of $\{X^n\}$ and verify the two conditions, as in the following two lemmas for the Lyapunov function and the minorization over a small set, respectively.

**Lemma A.2.** *Assume* $f$ *satisfies Assumption 3.1 and*

$$h \leq \frac{\mu \min\{\phi_i\}}{4 + 8L^2 + 32L^4}, \tag{35}$$

*where L is the Lipschitz constant defined in* (9). *Let the Lyapunov function be* $L(x) = |x - x^*|^2 + 1$, *then we have:*

$$\mathbb{E}\left(L(X^{m+1}) \big| \mathcal{F}^m\right) \leq \alpha_1 L(X^m) + \beta_1 \tag{36}$$

*with*

$$\alpha_1 = 1 - \mu h, \quad \beta_1 = (24 + 120L^2 + \mu)h.$$

**Lemma A.3.** *Under conditions of Lemma A.2, with* $L(x) = |x - x^*|^2 + 1$, *let* $\Xi$ *denote the transition kernel. Define the set* $C \subset \mathbb{R}^d$ *as in* (33), *for some* $\gamma \in (\alpha^{1/2}, 1)$. *Then there exists an* $\eta > 0$ *and a probability measure* $\mathcal{M}$ *with* $\mathcal{M}(C) = 1$, *such that*

$$\Xi^d(x, A) \geq \eta \mathcal{M}(A), \quad \forall A \in \mathcal{B}(\mathbb{R}^d), x \in C. \tag{37}$$

Lemma A.1 follows easily from these results.

*Proof of Lemma A.1.* It suffices to show $d$-step chain $\left\{X^{md}\right\}_{m=0}^{\infty}$ satisfies the conditions in Theorem A.1 with $L(x) = |x - x^*|^2 + 1$, $\alpha = \alpha_1^d$ and $\beta = d\beta_1$, and $\pi$ is induced by $p$. We apply (36) from Lemma A.2 iteratively, $d$ times, to obtain

$$\mathbb{E}\left(L\left(X^{(m+1)d}\right) \big| \mathcal{F}^{md}\right) \leq \alpha_1^d L\left(X^{md}\right) + d\beta_1,$$

which implies that $\left\{X^{md}\right\}_{m=0}^{\infty}$ satisfies Lyapunov condition in Theorem A.1 with $\alpha = \alpha_1^d$. Moreover, Lemma A.3 directly implies that the $d$-step transition kernel satisfies the minorization condition. Therefore, by Theorem A.1, we have

$$\sup_{A \in \mathcal{B}(\mathbb{R}^d)} \left| \Xi^{md}(z^0, A) - \pi(A) \right| \leq L(z^0) R r^{-m},$$

which concludes the proof of the lemma when we substitute $\pi(A) = \int_A p(x) \, \mathrm{d}x$. $\qquad \square$

*Proof of Lemma A.2.* We assume without loss of generality that $x^* = 0 \in \mathbb{R}^d$ (so that $L(x) = |x|^2 + 1$) and drop the filtration $\mathcal{F}^m$ in the formula for simplicity of notation. Then

$$\mathbb{E}\left(L\left(X^{m+1}\right)\right) = \sum_{i=1}^{d} \phi_i \mathbb{E}\left(L\left(X^{m+1}\right)\big|r^m = i\right). \tag{38}$$

Since

$$L\left(X^{m+1}\right) = |X^{m+1}|^2 + 1 = |X^m + (X^{m+1} - X^m)|^2 + 1$$
$$= L\left(X^m\right) + 2X^m(X^{m+1} - X^m) + |X^{m+1} - X^m|^2,$$

we have

$$\mathbb{E}\left(L\left(X^{m+1}\right)\big|r^m = i\right) = L\left(X^m\right) + 2\mathbb{E}\left[X_i^m\left(X_i^{m+1} - X_i^m\right)\big|r^m = i\right]$$
$$+ \mathbb{E}\left[\left(X_i^{m+1} - X_i^m\right)^2\big|r^m = i\right]. \tag{39}$$

To deal with second term and third term in (39), we first note that, under condition $r^m = i$:

$$X_i^{m+1} - X_i^m = -\int_{T^m}^{T^m + h_i} \partial_i f(X(s))\,\mathrm{d}s + \sqrt{2}\int_{T^m}^{T^m + h_i} \mathrm{d}B_s. \tag{40}$$

This means

$$2\mathbb{E}\left[X_i^m\left(X_i^{m+1} - X_i^m\right)\big|r^m = i\right]$$
$$= -2\mathbb{E}\left[X_i^m \int_{T^m}^{T^m + h_i} \partial_i f(X(s))\,\mathrm{d}s\,\bigg|r^m = i\right] \tag{41}$$
$$= -2h_i X_i^m \partial_i f(X^m) - 2\mathbb{E}\left[X_i^m \int_{T^m}^{T^m + h_i} (\partial_i f(X(s)) - \partial_i f(X^m))\,\mathrm{d}s\,\bigg|r^m = i\right].$$

We further bound the second term of (41):

$$\left|\mathbb{E}\left[X_i^m \int_{T^m}^{T^m + h_i} (\partial_i f(X(s)) - \partial_i f(X^m))\,\mathrm{d}s\,\bigg|r^m = i\right]\right|$$
$$\leq h_i \mathbb{E}\left[X_i^m \left(\sup_{T^m \leq t \leq T^m + h_i} |\partial_i f(X(t)) - \partial_i f(X^m)|\right)\bigg|r^m = i\right]$$
$$\overset{(\mathrm{I})}{\leq} 2h_i^2|X_i^m|^2 + 2\mathbb{E}\left(\sup_{T^m \leq t \leq T^m + h_i} |\partial_i f(X(t)) - \partial_i f(X^m)|^2\bigg|r^m = i\right) \tag{42}$$
$$\overset{(\mathrm{II})}{\leq} 2h_i^2|X_i^m|^2 + 2L_i^2\mathbb{E}\left(\sup_{T^m \leq t \leq T^m + h_i} |X_i(t) - X_i^m|^2\bigg|r^m = i\right)$$
$$\overset{(\mathrm{III})}{\leq} 2h_i^2|X_i^m|^2 + 16h_i^2 L_i^2|\partial_i f(X^m)|^2 + 60h_i L_i^2$$
$$\overset{(\mathrm{IV})}{\leq} (2 + 16L_i^4)h_i^2|X_i^m|^2 + 60h_i L_i^2,$$

where we used Young's inequality in (I), the Lipschitz condition in (II), Lemma A.4 below (specifically, inequality (45)) in (III), and the Lipschitz condition again in (IV). This, when substituted into (41), gives

$$2\mathbb{E}\left[X_i^m\left(X_i^{m+1} - X_i^m\right)\big|r^m = i\right] \leq -2h_i X_i^m \partial_i f(X^m) + (4 + 32L_i^4)h_i^2|X_i^m|^2 + 120h_i L_i^2.$$

To bound the third term in (39), again for the case $r^m = i$, we use (40) again for:

$$
\mathbb{E}\left[\left(X_i^{m+1} - X_i^m\right)^2 \middle| r^m = i\right]
$$

$$
= \mathbb{E}\left[\left(\int_{T^m}^{T^m + h_i} \partial_i f(X(s))\,\mathrm{d}s - \sqrt{2}\int_{T^m}^{T^m + h_i}\mathrm{d}B_s\right)^2 \middle| r^m = i\right]
$$

$$
\overset{(\mathrm{I})}{\leq} 2h_i^2 \mathbb{E}\left(\sup_{T^m \leq t \leq T^m + h_i}|\partial_i f(X(t))|^2 \middle| r^m = i\right) + 4\mathbb{E}\left(\left|\int_{T^m}^{T^m + h_i}\mathrm{d}B_s\right|^2 \middle| r^m = i\right) \tag{43}
$$

$$
= 2h_i^2 \mathbb{E}\left(\sup_{T^m \leq t \leq T^m + h_i}|\partial_i f(X(t))|^2 \middle| r^m = i\right) + 4h_i
$$

$$
\overset{(\mathrm{II})}{\leq} 8h_i^2|\partial_i f(X^m)|^2 + 88h_i^3 L_i^2 + 4h_i
$$

$$
\overset{(\mathrm{III})}{\leq} 8L_i^2 h_i^2 |X_i^m|^2 + 24h_i\,,
$$

where we used Young's inequality in (I), Lemma A.4 below (specifically, inequality (44)) in (II), and Lipschitz continuity in (III), together with $88h_i^2 L_i^2 \leq 20$ by (35).

Finally, we have

$$
\mathbb{E}\left(L\left(X^{m+1}\right)\middle| r^m = i\right) \leq L\left(X^m\right) - 2h_i X_i^m \partial_i f(X^m) + (4 + 8L_i^2 + 32L_i^4)h_i^2|X_i^m|^2 + (24 + 120L_i^2)h_i\,.
$$

By summing according to (38), and using (4) and $L_i \leq L$ for all $i = 1, 2, \ldots, d$, we obtain

$$
\mathbb{E}\left(L\left(X^{m+1}\right)\right) = \sum_{i=1}^{d}\phi_i \mathbb{E}\left(L\left(X^{m+1}\right)\middle| r^m = i\right)
$$

$$
\leq L\left(X^m\right) - 2h\left\langle X^m, \nabla f(X^m)\right\rangle + \frac{\left(4 + 8L^2 + 32L^4\right)h^2}{\min\{\phi_i\}}\left(L\left(X^m\right) - 1\right) + (24 + 120L^2)h\,.
$$

Finally, using $\left\langle X^m, \nabla f(X^m)\right\rangle \geq \mu(L\left(X^m\right) - 1)$ (from (10) with $x' = X^m$ and $x = x^* = 0$) and (35), we obtain (36). $\qquad\square$

*Proof of Lemma A.3.* To prove (37), we construct a new Markov process $\widetilde{X}^m$. Defining $\widetilde{X}^0 = x^0$, we obtain $\widetilde{X}^{m+1}$ from $\widetilde{X}^m$ by running the following process:

$$
\widetilde{T}^n = \sum_{i=1}^{n}h_i, \quad \widetilde{T}^0 = 0, \quad Z(0) = \widetilde{X}^m\,.
$$

Then for $\widetilde{T}^{n-1} \leq t \leq \widetilde{T}^n$ and $n \leq d$, let

$$
\begin{cases}
Z_n(t) = Z_n\left(\widetilde{T}^{n-1}\right) - \int_{\widetilde{T}^{n-1}}^{t}\partial_n f\left(Z(s)\right)\mathrm{d}s + \sqrt{2}\int_{\widetilde{T}^{n-1}}^{t}\mathrm{d}B_s\,, \\
Z_i(t) = Z_i\left(\widetilde{T}^{n-1}\right), \quad i \neq n\,,
\end{cases}
$$

and set $\widetilde{X}^{m+1} = Z\left(\widetilde{T}^d\right)$. Denote the transition kernel by $\Xi_{\mathrm{cyc}}$ (corresponding to one round of a cyclic version of the coordinate algorithm). We then have the following properties:

- For any $x \in C$ and $A \in \mathcal{B}(\mathbb{R}^d)$, we have

$$
\Xi^d(x, A) \geq \Pi_{i=1}^d \phi_i \Xi_{\mathrm{cyc}}(x, A) > 0\,.
$$

- $\Xi_{\mathrm{cyc}}$ possesses a positive jointly continuous density.

According to (Mattingly et al., 2002, Lemma 2.3), since $\Xi_{\mathrm{cyc}}$ has a positive jointly continuous density, there exists an $\eta' > 0$ and a probability measure $\mathcal{M}$ with $\mathcal{M}(C) = 1$, such that

$$
\Xi_{\mathrm{cyc}}(x, A) > \eta' \mathcal{M}(A), \quad \forall A \in \mathcal{B}\left(\mathbb{R}^d\right), x \in C\,,
$$

which implies

$$\Xi^d(x, A) \geq \Pi_{i=1}^d \phi_i \Xi_{\text{cyc}}(x, A) > \Pi_{i=1}^d \phi_i \eta' \mathcal{M}(A), \quad \forall A \in \mathcal{B}\left(\mathbb{R}^d\right), x \in C.$$

This proves (37) by setting $\eta = \Pi_{i=1}^d \phi_i \eta'$. $\qquad\square$

In the proof of Lemma A.2, we used several estimates in inequalities (42) and (43). We prove these estimates in the following lemma.

**Lemma A.4.** *Suppose that the assumptions of Lemma A.2 hold, and let $X_i$ evolve according to* (40). *Then we have the following bounds:*

$$\mathbb{E}\left(\sup_{T^m \leq t \leq T^m + h_i} |\partial_i f(X(t))|^2\right) \leq 4|\partial_i f(X^m)|^2 + 44h_i L_i^2, \tag{44}$$

$$\mathbb{E}\left(\sup_{T^m \leq t \leq T^m + h_i} |X_i(t) - X_i^m|^2\right) \leq 8h_i^2 |\partial_i f(X^m)|^2 + 30h_i. \tag{45}$$

*Proof.* To obtain (44), we have

$$\mathbb{E}\left(\sup_{T^m \leq t \leq T^m + h_i} |\partial_i f(X(t))|^2\right)$$

$$\leq \mathbb{E}\left[\sup_{T^m \leq t \leq T^m + h_i} \left(|\partial_i f(X^m)| + L_i |X_i(t) - X_i^m|\right)^2\right] \tag{46}$$

$$\leq 2|\partial_i f(X^m)|^2 + 2L_i^2 \mathbb{E}\left(\sup_{T^m \leq t \leq T^m + h_i} |X_i(t) - X_i^m|^2\right).$$

To bound the second term, we use (40) again:

$$\mathbb{E}\left(\sup_{T^m \leq t \leq T^m + h_i} |X_i(t) - X_i^m|^2\right)$$

$$= \mathbb{E}\left(\sup_{T^m \leq t \leq T^m + h_i} \left|\int_{T^m}^t \partial_i f\left(X(s)\right) \, \mathrm{d}s - \sqrt{2}\int_{T^m}^t \mathrm{d}B_s\right|^2\right) \tag{47}$$

$$\leq 2h_i^2 \mathbb{E}\left(\sup_{T^m \leq t \leq T^m + h_i} |\partial_i f(X(t))|^2\right) + 4\mathbb{E}\left(\sup_{T^m \leq t \leq T^m + h_i} \left|\int_{T^m}^t \mathrm{d}B_s\right|^2\right)$$

$$\leq 2h_i^2 \mathbb{E}\left(\sup_{T^m \leq t \leq T^m + h_i} |\partial_i f(X(t))|^2\right) + 16h_i,$$

where we use Young's inequality and

$$\mathbb{E}\left(\sup_{T^m \leq t \leq T^m + h_i} \left|\int_{T^m}^t \mathrm{d}B_s\right|^2\right) \leq 4\mathbb{E}\left(\left|\int_{T^m}^{T^m + h_i} \mathrm{d}B_s\right|^2\right) = 4h_i$$

by Doob's maximal inequality. By substituting (47) into (46), we obtain

$$\mathbb{E}\left(\sup_{T^m \leq t \leq T^m + h_i} |\partial_i f(X(t))|^2\right)$$

$$\leq 4h_i^2 L_i^2 \mathbb{E}\left(\sup_{T^m \leq t \leq T^m + h_i} |\partial_i f(X(t))|^2\right) + 2|\partial_i f(X^m)|^2 + 32h_i L_i^2.$$

Using $h_i L_i \leq \frac{1}{4}$, we move the first term on the right to the left to obtain

$$\frac{3}{4}\mathbb{E}\left(\sup_{T^m \leq t \leq T^m + h_i} |\partial_i f(X(t))|^2\right) \leq 2|\partial_i f(X^m)|^2 + 32h_i L_i^2,$$

leading to (44). Then we obtain (45) by plugging this in (47) and using the fact that $88h_i^3 L_i^2 < 14h_i$ by (35). $\qquad\square$

# B   PROOF OF THEOREM 4.1

The proof of this theorem requires us to design a reference solution to explicitly bound $W(q_m, p)$. Let $\tilde{x}^0$ be a random vector drawn from target distribution induced by $p$, so that $W_2^2(q_0, p) = \mathbb{E}|x^0 - \tilde{x}^0|^2$. We then require $\tilde{x}$ to solve the following SDE: for $t \in (T^m, T^{m+1}]$, with $T^m$ defined in (6):

$$\begin{cases} \tilde{x}_{r^m}(t) = \tilde{x}_{r^m}(T^m) - \int_{T^m}^t \partial_{r^m} f(\tilde{x}(s)) \, \mathrm{d}s + \sqrt{2} \int_{T^m}^t \mathrm{d}B_s \,, \\ \tilde{x}_i(t) = \tilde{x}_i(T^m), \quad i \neq r^m \,. \end{cases} \tag{48}$$

If we use the same Brownian motion as in (5), we have

$$\tilde{x}^{m+1} = \tilde{x}^m + \left[ -\int_{T^m}^{T^{m+1}} \partial_{r^m} f(\tilde{x}(s)) \, \mathrm{d}s + \sqrt{2h_{r^m}} \xi^m \right] e_{r^m} \,, \tag{49}$$

where $e_{r^m}$ is the unit vector in $r^m$ direction. Since the $r^m$-th marginal distribution of $\tilde{x}(t)$ is preserved in each time step according to (48), the whole distribution of $\tilde{x}(t)$ is preserved to be $p$ for all $t$. Therefore, by the definition $W_m = W(q_m, p)$, we have

$$W_m^2 \leq \mathbb{E}|\Delta^m|^2 = \mathbb{E}|x^m - \tilde{x}^m|^2 \,,$$

where

$$\Delta^m := \tilde{x}^m - x^m \,. \tag{50}$$

This means bounding $W_m$ amounts to evaluating $\mathbb{E}|\Delta^m|^2$. Under Assumption 3.1, we have the following result.

**Proposition B.1.** *Suppose the assumptions of Theorem 4.1 are satisfied and let $\{x^m\}$, $\{\tilde{x}^m\}$, and $\{\Delta^m\}$ be defined in (5), (48), and (50), respectively. Then, we have*

$$\mathbb{E}|\Delta^{m+1}|^2 \leq \left(1 - \frac{h\mu}{2}\right) \mathbb{E}|\Delta^m|^2 + \frac{10h^2}{\mu} \sum_{i=1}^d \frac{L_i^2}{\phi_i} \,. \tag{51}$$

The proof of this result appears in Appendix B.1. The proof for Theorem 4.1 is now immediate.

*Proof of Theorem 4.1.* By iterating (51), we obtain

$$\mathbb{E}|\Delta^m|^2 \leq \left(1 - \frac{h\mu}{2}\right)^m \mathbb{E}|\Delta^0|^2 + \frac{20h}{\mu^2} \sum_{i=1}^d \frac{L_i^2}{\phi_i} \,,$$

and since $h\mu/2 \in (0, 1)$, we have

$$\mathbb{E}|\Delta^m|^2 \leq \exp\left(-\frac{\mu h m}{2}\right) \mathbb{E}|\Delta^0|^2 + \frac{20h}{\mu^2} \sum_{i=1}^d \frac{L_i^2}{\phi_i} \,. \tag{52}$$

By construction, we have $W^2(q_0, p) = \mathbb{E}|\Delta^0|^2$ and $W^2(q_m, p) \leq \mathbb{E}|\Delta^m|^2$. By taking the square root of both sides and using $a^2 \leq b^2 + c^2 \Rightarrow a \leq b + c$ for any nonnegative $a$, $b$, and $c$, we arrive at (20). $\qquad\square$

The proof for Corollary 4.1 is also obvious.

*Proof of Corollary 4.1.* To ensure that $W_m \leq \epsilon$, we set the two terms on the right hand side of (20) to be smaller than $\epsilon/2$, which implies that

$$h = O\left(\frac{\mu^2 \epsilon^2}{100 \sum_{i=1}^d \frac{L_i^2}{\phi_i(\alpha)}}\right) \quad \text{and} \quad m \geq \frac{4}{\mu h} \log\left(\frac{2W_0}{\epsilon}\right) \,. \tag{53}$$

By using the definition of $\phi_i(\alpha)$ according to (21), we obtain

$$\sum_{i=1}^d \frac{L_i^2}{\phi_i(\alpha)} = \left(\sum_{i=1}^d \frac{L_i^2}{L_i^\alpha}\right) \left(\sum_{j=1}^d L_j^\alpha\right) = \mu^2 K_{2-\alpha} K_\alpha \,,$$

which implies that $m = \widetilde{O}\left((K_{2-\alpha}K_\alpha)/(\mu\epsilon^2)\right)$. Furthermore, $\alpha = 1$ gives the optimal cost, because:

$$K_{2-\alpha}K_\alpha = \left(\sum \kappa_i^\alpha\right)\left(\sum \kappa_i^{2-\alpha}\right) \geq \left(\sum_i \kappa_i\right)^2 = K_1^2,$$

due to Hölder's inequality. □

## B.1 Proof of Proposition B.1

We prove the Proposition by means of the following lemma.

**Lemma B.1.** *Under the conditions of Proposition B.1, for $m \geq 0$ and $i = 1, 2, \ldots, d$, we have*

$$
\mathbb{E}|\Delta_i^{m+1}|^2 \leq \left(1 + h\mu + \frac{h^2\mu^2}{\phi_i}\right)\mathbb{E}|\Delta_i^m|^2 - 2h\mathbb{E}\left[\Delta_i^m\left(\partial_i f(\tilde{x}^m) - \partial_i f(x^m)\right)\right]
$$
$$
+ \frac{3h^2}{\phi_i}\mathbb{E}\left|\partial_i f(\tilde{x}^m) - \partial_i f(x^m)\right|^2 + \left(\frac{2h^3 L_i^3}{\mu\phi_i^2} + \frac{8h^2 L_i^2}{\mu\phi_i}\right). \tag{54}
$$

*Proof.* In the $m$-th time step, we have

$$\mathbb{P}(r^m = i) = \phi_i, \quad \mathbb{P}(r^m \neq i) = 1 - \phi_i,$$

so that

$$
\mathbb{E}|\Delta_i^{m+1}|^2 = \phi_i\mathbb{E}\left(|\Delta_i^{m+1}|^2 \mid r^m = i\right) + (1 - \phi_i)\mathbb{E}\left(|\Delta_i^{m+1}|^2 \mid r^m \neq i\right)
$$
$$
= \phi_i\mathbb{E}\left(|\Delta_i^{m+1}|^2 \mid r^m = i\right) + (1 - \phi_i)\mathbb{E}|\Delta_i^m|^2. \tag{55}
$$

We now analyze the first term on the right hand side under condition $r^m = i$. By definition of $\Delta_i^{m+1}$, we have

$$
\begin{aligned}
\Delta_i^{m+1} &= \Delta_i^m + (\tilde{x}_i^{m+1} - \tilde{x}_i^m) - (x_i^{m+1} - x_i^m) \\
&= \Delta_i^m + \left(-\int_{T^m}^{T^m + h_i} \partial_i f(\tilde{x}(s))\,\mathrm{d}s + \sqrt{2h_i}\xi_m\right) - \left(-\int_{T^m}^{T^m + h_i} \partial_i f(x^m)\,\mathrm{d}s + \sqrt{2h_i}\xi_m\right) \\
&= \Delta_i^m - \int_{T^m}^{T^m + h_i} \left(\partial_i f(\tilde{x}(s)) - \partial_i f(x^m)\right)\,\mathrm{d}s \\
&= \Delta_i^m - \int_{T^m}^{T^m + h_i} \left(\partial_i f(\tilde{x}(s)) - \partial_i f(\tilde{x}^m) + \partial_i f(\tilde{x}^m) - \partial_i f(x^m)\right)\,\mathrm{d}s \\
&= \Delta_i^m - h_i\left(\partial_i f(\tilde{x}^m) - \partial_i f(x^m)\right) - \int_{T^m}^{T^m + h_i} \left(\partial_i f(\tilde{x}(s)) - \partial_i f(\tilde{x}^m)\right)\,\mathrm{d}s \\
&= \Delta_i^m - h_i\left(\partial_i f(\tilde{x}^m) - \partial_i f(x^m)\right) - V^m,
\end{aligned} \tag{56}
$$

where we have defined

$$V^m := \int_{T^m}^{T^m + h_i} \left(\partial_i f(\tilde{x}(s)) - \partial_i f(\tilde{x}^m)\right)\,\mathrm{d}s. \tag{57}$$

By Young's inequality, we have

$$
\begin{aligned}
&\mathbb{E}\left(|\Delta_i^{m+1}|^2 \mid r^m = i\right) \\
&= \mathbb{E}\left(|\Delta_i^{m+1} + V^m - V^m|^2 \mid r^m = i\right) \\
&\leq (1 + a)\mathbb{E}\left(|\Delta_i^{m+1} + V^m|^2 \mid r^m = i\right) + \left(1 + \frac{1}{a}\right)\mathbb{E}\left(|V^m|^2 \mid r^m = i\right), \tag{58}
\end{aligned}
$$

where $a > 0$ is a parameter to be specified later.

For the first term on the right hand side of (58), we have

$$
\begin{aligned}
&\mathbb{E}\left(|\Delta_i^{m+1} + V^m|^2 \mid r^m = i\right) \\
&= \mathbb{E}|\Delta_i^m - h_i\left(\partial_i f(\tilde{x}^m) - \partial_i f(x^m)\right)|^2 \\
&= \mathbb{E}|\Delta_i^m|^2 - 2h_i\mathbb{E}\left[\Delta_i^m\left(\partial_i f(\tilde{x}^m) - \partial_i f(x^m)\right)\right] + h_i^2\mathbb{E}\left|\partial_i f(\tilde{x}^m) - \partial_i f(x^m)\right|^2. \tag{59}
\end{aligned}
$$

Note that the second term will essentially become the second line in (54), and the third term will become the third line in (54) (upon the proper choice of $a$). For very small $h$, this term is negligible.

For the second term on the right-hand side of (58), we recall the definition (57) and obtain

$$
\begin{aligned}
\mathbb{E}\left(|V^m|^2\big|r^m=i\right) &\stackrel{\text{(I)}}{\leq} h_i \int_{T^m}^{T^m+h_i} \mathbb{E}\left(|\partial_i f(\tilde{x}(s)) - \partial_i f(\tilde{x}^m)|^2\big|r^m=i\right) \,\mathrm{d}s \\
&\stackrel{\text{(II)}}{\leq} h_i L_i^2 \int_{T^m}^{T^m+h_i} \mathbb{E}\left(|\tilde{x}(s) - \tilde{x}^m|^2\big|r^m=i\right) \,\mathrm{d}s \\
&= h_i L_i^2 \int_{T^m}^{T^m+h_i} \mathbb{E}\left(\left|\int_{T^m}^s \partial_i f(\tilde{x}(t))\,\mathrm{d}t + \sqrt{2}(B_s - B_{T^m})\right|^2\bigg|r^m=i\right) \,\mathrm{d}s \\
&\stackrel{\text{(III)}}{\leq} 2h_i^2 L_i^2 \int_{T^m}^{T^m+h_i} \int_{T^m}^s \mathbb{E}\left(|\partial_i f(\tilde{x}(t))|^2\big|r^m=i\right) \,\mathrm{d}t\,\mathrm{d}s \\
&\quad + 4h_i^2 L_i^2 \int_{T^m}^{T^m+h_i} \mathbb{E}|\xi^m|^2 \,\mathrm{d}s \\
&\stackrel{\text{(IV)}}{=} h_i^4 L_i^2 \mathbb{E}\left(|\partial_i f(\tilde{x}^m)|^2\right) + 4h_i^3 L_i^2 \\
&\stackrel{\text{(V)}}{=} h_i^4 L_i^2 \mathbb{E}_p|\partial_i f|^2 + 4h_i^3 L_i^2 \stackrel{\text{(VI)}}{\leq} h_i^4 L_i^3 + 4h_i^3 L_i^2\,,
\end{aligned}
\tag{60}
$$

where (II) comes from $L$-Lipschitz condition (11), (I) and (III) come from the use of Young's inequality and Jensen's inequality when we move the $|\cdot|^2$ from outside to inside of the integral, and (IV) and (V) hold true because $\tilde{x}(t) \sim p$ for all $t$. In (VI) we use $\mathbb{E}_p|\partial_i f|^2 \leq L_i$ using (Dalalyan and Karagulyan, 2019, Lemma 3).

By substituting (59) and (60) into the right hand side of (58), we obtain

$$
\begin{aligned}
\mathbb{E}&\left(|\Delta_i^{m+1}|^2 \mid r^m=i\right) \\
&\leq (1+a)\mathbb{E}|\Delta_i^m|^2 - 2h_i(1+a)\mathbb{E}\left[\Delta_i^m\left(\partial_i f(\tilde{x}^m) - \partial_i f(x^m)\right)\right] \\
&\quad + h_i^2(1+a)\mathbb{E}\left|\partial_i f(\tilde{x}^m) - \partial_i f(x^m)\right|^2 + \left(1+\frac{1}{a}\right)\left(h_i^4 L_i^3 + 4h_i^3 L_i^2\right)\,.
\end{aligned}
\tag{61}
$$

By substituting (61) into (55), we have

$$
\begin{aligned}
\mathbb{E}|\Delta_i^{m+1}|^2 &\leq (1+a\phi_i)\,\mathbb{E}|\Delta_i^m|^2 - 2(1+a)h\mathbb{E}\left[\Delta_i^m\left(\partial_i f(\tilde{x}^m) - \partial_i f(x^m)\right)\right] \\
&\quad + \frac{(1+a)h^2}{\phi_i}\mathbb{E}\left|\partial_i f(\tilde{x}^m) - \partial_i f(x^m)\right|^2 + \left(1+\frac{1}{a}\right)\left(\frac{h^4 L_i^3}{\phi_i^3} + \frac{4h^3 L_i^2}{\phi_i^2}\right)\,,
\end{aligned}
\tag{62}
$$

where we have used $h_i\phi_i = h$.

Now, we need to choose a value of $a > 0$ appropriate to establish (54). By comparing the two formulas, we see the need to set

$$
a\phi_i = h\mu \quad \Rightarrow \quad a = h_i\mu = \frac{h\mu}{\phi_i} \leq 1\,.
$$

since $h \leq \min\{\phi_i\}/\mu$. It follows that $1 + \frac{1}{a} \leq \frac{2\phi_i}{h\mu}$. By substituting into (62), we obtain

$$
\begin{aligned}
\mathbb{E}|\Delta_i^{m+1}|^2 &\leq (1+h\mu)\,\mathbb{E}|\Delta_i^m|^2 - 2h\mathbb{E}\left[\Delta_i^m\left(\partial_i f(\tilde{x}^m) - \partial_i f(x^m)\right)\right] \\
&\quad - \frac{2h^2\mu}{\phi_i}\mathbb{E}\left[\Delta_i^m\left(\partial_i f(\tilde{x}^m) - \partial_i f(x^m)\right)\right] + \frac{2h^2}{\phi_i}\mathbb{E}\left|\partial_i f(\tilde{x}^m) - \partial_i f(x^m)\right|^2 \\
&\quad + \left(\frac{2h^3 L_i^3}{\mu\phi_i^2} + \frac{8h^2 L_i^2}{\mu\phi_i}\right)\,.
\end{aligned}
\tag{63}
$$

We conclude the lemma by using the following Cauchy-Schwartz inequality to control the third term on the right hand side of this expression:

$$
-\frac{2h^2\mu}{\phi_i}\mathbb{E}\left[\Delta_i^m\left(\partial_i f(\tilde{x}^m) - \partial_i f(x^m)\right)\right] \leq \frac{h^2\mu^2}{\phi_i}\mathbb{E}|\Delta_i^m|^2 + \frac{h^2}{\phi_i}\mathbb{E}|\partial_i f(\tilde{x}^m) - \partial_i f(x^m)|^2\,. \qquad \square
$$

Proposition B.1 is obtained by simply summing all components in the lemma.

*Proof of Proposion B.1.* Noting

$$\mathbb{E}|\Delta^{m+1}|^2 = \sum_{i=1}^{d} \mathbb{E}|\Delta_i^{m+1}|^2,$$

we bound the right hand side by (54) and get

$$\mathbb{E}|\Delta^{m+1}|^2 \leq \left(1 + h\mu + \frac{h^2\mu^2}{\min\{\phi_i\}}\right)\mathbb{E}|\Delta^m|^2 - 2h\mathbb{E}\left\langle \Delta^m, \nabla f(\tilde{x}^m) - \nabla f(x^m)\right\rangle$$

$$+ \frac{3h^2}{\min\{\phi_i\}}\mathbb{E}\left|\nabla f(\tilde{x}^m) - \nabla f(x^m)\right|^2 + \left(\frac{2h^3}{\mu}\sum_{i=1}^{d}\frac{L_i^3}{\phi_i^2} + \frac{8h^2}{\mu}\sum_{i=1}^{d}\frac{L_i^2}{\phi_i}\right). \tag{64}$$

The second and third terms on the right-hand side can be bounded in terms of $\mathbb{E}|\Delta^m|^2$:

- By convexity, we have

$$\mathbb{E}\left\langle \Delta^m, \nabla f(\tilde{x}^m) - \nabla f(x^m)\right\rangle \geq \mu\mathbb{E}|\Delta^m|^2. \tag{65}$$

- As the gradient is $L$-Lipschitz, we have

$$\mathbb{E}\left|\nabla f(\tilde{x}^m) - \nabla f(x^m)\right|^2 \leq L^2\mathbb{E}|\Delta^m|^2. \tag{66}$$

By substituting (65) and (66) into (64) and using $\mu \leq L$, we obtain

$$\mathbb{E}|\Delta^{m+1}|^2 \leq \left(1 - h\mu + \frac{4h^2L^2}{\min\{\phi_i\}}\right)\mathbb{E}|\Delta^m|^2 + \left(\frac{2h^3}{\mu}\sum_{i=1}^{d}\frac{L_i^3}{\phi_i^2} + \frac{8h^2}{\mu}\sum_{i=1}^{d}\frac{L_i^2}{\phi_i}\right). \tag{67}$$

If we take $h$ sufficiently small, the coefficient in front of $\mathbb{E}|\Delta^m|^2$ is strictly smaller than 1, ensuring the decay of the error. Indeed, by setting $h \leq \frac{\mu\min\{\phi_i\}}{8L^2}$, we have

$$\frac{4h^2L^2}{\min\{\phi_i\}} \leq \frac{h\mu}{2}, \quad \text{and} \quad \frac{hL_i}{\phi_i} \leq \frac{\mu}{8L} \leq 1,$$

which leads to the iteration formula (51). □

## C  PROOF OF THEOREM 4.2

Theorem 4.2 is based on the following proposition.

**Proposition C.1.** *Suppose the assumptions of Theorem 4.2 and let $\{x^m\}$, $\{\tilde{x}^m\}$, and $\{\Delta_m\}$ be defined as in* (5), (48), *and* (50), *respectively. Then we have*

$$\mathbb{E}|\Delta^{m+1}|^2 \leq \left(1 - \frac{h\mu}{2}\right)\mathbb{E}|\Delta^m|^2 + \frac{4h^3}{\mu}\sum_{i=1}^{d}\frac{\left(L_i^3 + H_i^2\right)}{\phi_i^2}. \tag{68}$$

We prove this result in Appendix C.1. The proof of the theorem is now immediate.

*Proof of Theorem 4.2.* Use (68) iteratively, we have

$$\mathbb{E}|\Delta^{m+1}|^2 \leq \left(1 - \frac{h\mu}{2}\right)^m \mathbb{E}|\Delta^0|^2 + \frac{8h^2}{\mu^2}\sum_{i=1}^{d}\frac{\left(L_i^3 + H_i^2\right)}{\phi_i^2}$$

$$\leq \exp\left(-\frac{\mu hm}{2}\right)\mathbb{E}|\Delta^0|^2 + \frac{8h^2}{\mu^2}\sum_{i=1}^{d}\frac{\left(L_i^3 + H_i^2\right)}{\phi_i^2}.$$

Using $W^2(q_0, p) = \mathbb{E}|\Delta^0|^2$ and $W^2(q_m, p) \leq \mathbb{E}|\Delta^m|^2$, we take the square root on both sides, we obtain (23). □

The proof of Corollary 4.2 is also immediate.

*Proof of Corollary 4.2.* Use (23), to ensure $W_m \leq \epsilon$, we set two terms on the right hand side of (23) to be smaller than $\epsilon/2$, which implies that

$$
h = O\left(\frac{\epsilon\mu}{\sqrt{\sum_{i=1}^{d} \frac{(L_i^3 + H_i^2)}{\phi_i^2}}}\right), \quad m \geq \frac{4}{\mu h}\log\left(\frac{2W_0}{\epsilon}\right). \tag{69}
$$

To find optimal choice of $\phi_i$, we need to minimize

$$
\sum_{i=1}^{d} \frac{(L_i^3 + H_i^2)}{\phi_i^2}
$$

under constraint $\sum_i^d \phi_i = 1$ and $\phi_i > 0$. Introducing a Lagrange multiplier $\lambda \in \mathbb{R}$, define the Lagrangian function as follows:

$$
F(\phi_1, \phi_2, \ldots, \phi_d, \lambda) = \sum_{i=1}^{d} \frac{(L_i^3 + H_i^2)}{\phi_i^2} + \lambda\left(\sum_{i=1}^{d} \phi_i - 1\right).
$$

By setting $\partial F/\partial \phi_i = 0$ for all $i$, and substituting into the constraint $\sum_i^d \phi_i = 1$ to find the appropriate value of $\lambda$, we find that the optimal $(\phi_1, \phi_2, \ldots, \phi_d)$ satisfies

$$
\phi_i = \frac{(L_i^3 + H_i^2)^{1/3}}{\sum_{i=1}^{d} (L_i^3 + H_i^2)^{1/3}}, \quad i = 1, 2, \ldots, d.
$$

By substituting into (69), we obtain (24). □

## C.1 PROOF OF PROPOSITION C.1

The strategy of the proof for this proposition is almost identical to that of the previous section. The reference solution $\tilde{x}$ is defined as in (48). We will use the following lemma:

**Lemma C.1.** *Under the conditions of Proposition C.1, for $m \geq 0$ and $i = 1, 2, \ldots, d$, we have*

$$
\mathbb{E}|\Delta_i^{m+1}|^2 \leq \left(1 + h\mu + \frac{h^2\mu^2}{\phi_i}\right)\mathbb{E}|\Delta_i^m|^2 - 2h\mathbb{E}\left[\Delta_i^m\left(\partial_i f(\tilde{x}^m) - \partial_i f(x^m)\right)\right]
$$
$$
+ \frac{3h^2}{\phi_i}\mathbb{E}\left|\partial_i f(\tilde{x}^m) - \partial_i f(x^m)\right|^2 + \frac{4h^3\left(L_i^3 + H_i^2\right)}{\phi_i^2\mu}. \tag{70}
$$

*Proof.* In the $m$-th time step, we have

$$
\mathbb{P}(r^m = i) = \phi_i, \quad \mathbb{P}(r^m \neq i) = 1 - \phi_i,
$$

meaning that

$$
\mathbb{E}|\Delta_i^{m+1}|^2 = \phi_i \mathbb{E}\left(|\Delta_i^{m+1}|^2 \mid r^m = i\right) + (1 - \phi_i)\mathbb{E}\left(|\Delta_i^{m+1}|^2 \mid r^m \neq i\right)
$$
$$
= \phi_i \mathbb{E}\left(|\Delta_i^{m+1}|^2 \mid r^m = i\right) + (1 - \phi_i)\mathbb{E}|\Delta_i^m|^2. \tag{71}
$$

To bound the first term in (55) we use the definition of $\Delta_i^{m+1}$. Under the condition $r^m = i$, we have, with the same derivation as in (56):

$$
\Delta_i^{m+1} = \Delta_i^m - h_i\left(\partial_i f(\tilde{x}^m) - \partial_i f(x^m)\right) - \int_{T^m}^{T^m + h_i}\left(\partial_i f(\tilde{x}(s)) - \partial_i f(\tilde{x}^m)\right)\,\mathrm{d}s
$$
$$
= \Delta_i^m - h_i\left(\partial_i f(\tilde{x}^m) - \partial_i f(x^m)\right) - V^m, \tag{72}
$$

where we denoted $V^m = \int_{T^m}^{T^m + h_i}\left(\partial_i f(\tilde{x}(s)) - \partial_i f(\tilde{x}^m)\right)\,\mathrm{d}s$.

However, different from (60), since $f$ has higher regularity, we can find a tighter bound for the integral. Denote

$$U^m = \int_{T^m}^{T^m+h_i} \left( \partial_i f(\tilde{x}(s)) - \partial_i f(\tilde{x}^m) - \sqrt{2} \int_{T^m}^{s} \partial_{ii} f(\tilde{x}(z)) \, \mathrm{d}B_z \right) \mathrm{d}s \tag{73}$$

and

$$\Phi^m = \sqrt{2} \int_{T^m}^{T^m+h_i} \int_{T^m}^{s} \partial_{ii} f(\tilde{x}(z)) \, \mathrm{d}B_z \, \mathrm{d}s. \tag{74}$$

Then (72) can be written as

$$\Delta_i^{m+1} = \Delta_i^m - h_i \left( \partial_i f(\tilde{x}^m) - \partial_i f(x^m) \right) - \Phi^m - U^m, \tag{75}$$

which implies, according to Young's inequality, that, for any $a$:

$$\mathbb{E} \left( |\Delta_i^{m+1}|^2 \big| r^m = i \right) = \mathbb{E} \left( |\Delta_i^{m+1} + U^m - U^m|^2 \big| r^m = i \right)$$
$$\leq (1+a) \mathbb{E} \left( |\Delta_i^{m+1} + U^m|^2 \big| r^m = i \right) + \left( 1 + \frac{1}{a} \right) \mathbb{E} \left( |U^m|^2 \big| r^m = i \right). \tag{76}$$

Both terms on the right-hand side of (76) are small. We now control the first term. Plug in the definition (75), we have:

$$\mathbb{E} \left( |\Delta_i^{m+1} + U^m|^2 \mid r^m = i \right) = \mathbb{E} \left( |\Delta_i^m - h_i \left( \partial_i f(\tilde{x}^m) - \partial_i f(x^m) \right) - \Phi^m|^2 \big| r^m = i \right). \tag{77}$$

Noting that

$$\mathbb{E} \left( \left( \Delta_i^m - h_i \left( \partial_i f(\tilde{x}^m) - \partial_i f(x^m) \right) \right) \cdot \Phi^m \right)$$
$$= \sqrt{2} \int_{T^m}^{T^m+h_i} \mathbb{E} \left[ \int_{T^m}^{s} \left( \Delta_i^m - h_i \left( \partial_i f(\tilde{x}^m) - \partial_i f(x^m) \right) \right) \cdot \partial_{ii} f(\tilde{x}(z)) \, \mathrm{d}B_z \right] \mathrm{d}s = 0$$

because

$$\mathbb{E} \left[ \int_{T^m}^{s} \left( \Delta_i^m - h_i \left( \partial_i f(\tilde{x}^m) - \partial_i f(x^m) \right) \right) \cdot \partial_{ii} f(\tilde{x}(z)) \, \mathrm{d}B_z \right] = 0,$$

according to the property of Itô's integral, we can discard the cross terms with $\Phi^m$ in (77) to obtain

$$\mathbb{E} \left( |\Delta_i^{m+1} + U^m|^2 \mid r^m = i \right) = \mathbb{E} |\Delta_i^m|^2 - 2h_i \mathbb{E} \left[ \Delta_i^m \left( \partial_i f(\tilde{x}^m) - \partial_i f(x^m) \right) \right]$$
$$+ h_i^2 \mathbb{E} \left| \partial_i f(\tilde{x}^m) - \partial_i f(x^m) \right|^2 + \mathbb{E} \left( |\Phi^m|^2 \big| r^m = i \right). \tag{78}$$

For the last term of (78), we have the following control:

$$\mathbb{E} \left( |\Phi^m|^2 \big| r^m = i \right) = \mathbb{E} \left( 2 \left| \int_{T^m}^{T^m+h_i} \int_{T^m}^{s} \partial_{ii} f(\tilde{x}(z)) \, \mathrm{d}B_z \, \mathrm{d}s \right|^2 \Big| r^m = i \right)$$
$$\overset{(I)}{\leq} 2\mathbb{E} \left[ \left( \int_{T^m}^{T^m+h_i} \mathrm{d}s \right) \left( \int_{T^m}^{T^m+h_i} \left| \int_{T^m}^{s} \partial_{ii} f(\tilde{x}(z)) \, \mathrm{d}B_z \right|^2 \mathrm{d}s \right) \Big| r^m = i \right]$$
$$\leq 2h_i \int_{T^m}^{T^m+h_i} \mathbb{E} \left( \left| \int_{T^m}^{s} \partial_{ii} f(\tilde{x}(z)) \, \mathrm{d}B_z \right|^2 \Big| r^m = i \right) \mathrm{d}s$$
$$\overset{(II)}{=} 2h_i \int_{T^m}^{T^m+h_i} \int_{T^m}^{s} \mathbb{E} \left( |\partial_{ii} f(\tilde{x}(z))|^2 \big| r^m = i \right) \mathrm{d}z \, \mathrm{d}s$$
$$\overset{(III)}{=} h_i^3 \mathbb{E}_p |\partial_{ii} f|^2 = h_i^3 L_i^2,$$

where we use Hölder's inequality in I and $\tilde{x}(t) \sim p$ for all $t$ in III. In II, we use the following property of Itô's integral:

$$\mathbb{E} \left( \left| \int_{T^m}^{s} \partial_{ii} f(\tilde{x}(z)) \, \mathrm{d}B_z \right|^2 \Big| r^m = i \right) = \int_{T^m}^{s} \mathbb{E} \left( |\partial_{ii} f(\tilde{x}(z))|^2 \big| r^m = i \right) \mathrm{d}z.$$

By substituting into (78), we obtain

$$
\mathbb{E}\left(|\Delta_i^{m+1} + U^m|^2 \mid r^m = i\right) \leq \mathbb{E}|\Delta_i^m|^2 - 2h_i \mathbb{E}\left[\Delta_i^m \left(\partial_i f(\tilde{x}^m) - \partial_i f(x^m)\right)\right]
$$
$$
+ h_i^2 \mathbb{E}\left|\partial_i f(\tilde{x}^m) - \partial_i f(x^m)\right|^2 + h_i^3 L_i^2 \tag{79}
$$

To bound the second term on the right-hand side of (76), we first note that $f$ is three times continuously differentiable, and (15) implies $\|\partial_{iii} f\|_\infty \leq H_i$. Take $\mathrm{d}t$ on both sides of (48), under condition $r^m = i$, we first have

$$
\mathrm{d}\tilde{x}_i(t) = -\partial_i f(\tilde{x}(s))\,\mathrm{d}s + \sqrt{2}\,\mathrm{d}B_s \,. \tag{80}
$$

According to Itô's formula, we obtain

$$
\partial_i f(\tilde{x}(t)) - \partial_i f(\tilde{x}^m) = \int_{T^m}^t \partial_{ii} f(\tilde{x}(s))\,\mathrm{d}\tilde{x}_i(s) + \int_{T^m}^t \partial_{iii} f(\tilde{x}(s))\,\mathrm{d}s \,. \tag{81}
$$

Substituting (80) into (81), we have

$$
\partial_i f(\tilde{x}(t)) - \partial_i f(\tilde{x}^m) - \sqrt{2} \int_{T_m}^t \partial_{ii} f(\tilde{x}(s))\,\mathrm{d}B_s
$$
$$
= \int_{T^m}^t -\partial_{ii} f(\tilde{x}(s))\partial_i f(\tilde{x}(s)) + \partial_{iii} f(\tilde{x}(s))\,\mathrm{d}s \,. \tag{82}
$$

By substituting into (73), we obtain

$$
\mathbb{E}\left(|U^m|^2 \mid r^m = i\right)
$$
$$
\overset{\text{(I)}}{\leq} h_i \int_{T^m}^{T^m + h_i} \mathbb{E}\left(\left|\partial_i f(\tilde{x}(s)) - \partial_i f(\tilde{x}^m) - \sqrt{2}\int_{T^m}^s \partial_{ii} f(\tilde{x}(z))\,\mathrm{d}B_r\right|^2 \middle| r^m = i\right)\,\mathrm{d}s
$$
$$
\overset{\text{(II)}}{=} h_i \int_{T^m}^{T^m + h_i} \mathbb{E}\left(\left|\int_{T^m}^s \left(-\partial_{ii} f(\tilde{x}(z))\partial_i f(\tilde{x}(z)) + \partial_{iii} f(\tilde{x}(z))\right)\,\mathrm{d}z\right|^2 \middle| r^m = i\right)\,\mathrm{d}s
$$
$$
\overset{\text{(III)}}{\leq} h_i^2 \int_{T^m}^{T^m + h_i} \int_{T^m}^s \mathbb{E}\left(\left|\partial_{ii} f(\tilde{x}(z))\partial_i f(\tilde{x}(z)) + \partial_{iii} f(\tilde{x}(z))\right|^2 \middle| r^m = i\right)\,\mathrm{d}z\,\mathrm{d}s
$$
$$
\overset{\text{(IV)}}{\leq} 2h_i^2 \int_{T^m}^{T^m + h_i} \int_{T^m}^s \mathbb{E}\left(\left|\partial_{ii} f(\tilde{x}(z))\partial_i f(\tilde{x}(z))\right|^2 \middle| r^m = i\right)\,\mathrm{d}z\,\mathrm{d}s
$$
$$
+ 2h_i^2 \int_{T^m}^{T^m + h_i} \int_{T^m}^s \mathbb{E}\left(\left|\partial_{iii} f(\tilde{x}(z))\right|^2 \middle| r^m = i\right)\,\mathrm{d}z\,\mathrm{d}s
$$
$$
\overset{\text{(V)}}{\leq} h_i^4 \left(L_i^3 + H_i^2\right) \,. \tag{83}
$$

In the derivation, (II) comes from plugging in (82), and (I) and (III) come from the use of Jensen's inequality, (V) comes from the use of Lipschitz continuity in the first and the second derivative ((11) and (15) in particular), and the fact that $\tilde{x}(t) \sim p$ for all $t$. Note also $\mathbb{E}_p|\partial_i f|^2 \leq L_i$ by (Dalalyan and Karagulyan, 2019, Lemma 3).

By plugging (79) and (83) into (71) and (76), we obtain

$$
\mathbb{E}|\Delta_i^{m+1}|^2 \leq (1 + a\phi_i)\,\mathbb{E}|\Delta_i^m|^2 - 2(1+a)h\mathbb{E}\left[\Delta_i^m \left(\partial_i f(\tilde{x}^m) - \partial_i f(x^m)\right)\right]
$$
$$
+ \frac{(1+a)h^2}{\phi_i} \mathbb{E}\left|\partial_i f(\tilde{x}^m) - \partial_i f(x^m)\right|^2 + \frac{(1+a)h^3 L_i^2}{\phi_i^2} + \left(1 + \frac{1}{a}\right) \frac{h^4 \left(L_i^3 + H_i^2\right)}{\phi_i^3} \,, \tag{84}
$$

where we use $h_i \phi_i = h$. Comparing with (70), we need to set

$$
a = h_i \mu = \frac{h\mu}{\phi_i} < 1 \,,
$$

where we use $h < \frac{\mu \min\{\phi_i\}}{8L^2}$. This leads to $1 + \frac{1}{a} \leq \frac{2\phi_i}{h\mu}$. By substituting into (62), we obtain

$$
\mathbb{E}|\Delta_i^{m+1}|^2 \leq \left(1 + h\mu + \frac{h^2 \mu^2}{\phi_i}\right) \mathbb{E}|\Delta_i^m|^2 - 2h\mathbb{E}\left[\Delta_i^m \left(\partial_i f(\tilde{x}^m) - \partial_i f(x^m)\right)\right]
$$
$$
+ \frac{3h^2}{\phi_i} \mathbb{E}\left|\partial_i f(\tilde{x}^m) - \partial_i f(x^m)\right|^2 + \frac{2h^3 L_i^2}{\phi_i^2} + \frac{2h^3 \left(L_i^3 + H_i^2\right)}{\phi_i^2 \mu} \,.
$$

Noting $L_i/\mu > 1$, we conclude the lemma. $\qquad\square$

The proof of Proposition C.1 is obtained by summing up all components and applying Lemma C.1.

*Proof of Proposition C.1.* Noting that

$$\mathbb{E}|\Delta^{m+1}|^2 = \sum_{i=1}^d \mathbb{E}|\Delta_i^{m+1}|^2 \,,$$

we substitute using (70) to obtain

$$\mathbb{E}|\Delta^{m+1}|^2 \le \left(1 + h\mu + \frac{h^2\mu^2}{\min\{\phi_i\}}\right)\mathbb{E}|\Delta^m|^2 - 2h\mathbb{E}\langle\Delta^m, \nabla f(\tilde{x}^m) - \nabla f(x^m)\rangle$$
$$+ \frac{3h^2}{\min\{\phi_i\}}\mathbb{E}|\nabla f(\tilde{x}^m) - \nabla f(x^m)|^2 + \frac{4h^3}{\mu}\sum_{i=1}^d \frac{(L_i^3 + H_i^2)}{\phi_i^2}\,. \tag{85}$$

The second and third terms in the right-hand side of this bound can be controlled by $\mathbb{E}|\Delta^m|^2$, as follows. By convexity, we have

$$\mathbb{E}\langle\Delta^m, \nabla f(\tilde{x}^m) - \nabla f(x^m)\rangle \ge \mu\mathbb{E}|\Delta^m|^2\,. \tag{86}$$

By the $L$-Lipschitz property, we have

$$\mathbb{E}|\nabla f(\tilde{x}^m) - \nabla f(x^m)|^2 \le L^2\mathbb{E}|\Delta^m|^2\,. \tag{87}$$

By substituting (86) and (87) into (64), and using $\mu < L$, we have

$$\mathbb{E}|\Delta^{m+1}|^2 \le \left(1 - h\mu + \frac{4h^2L^2}{\min\{\phi_i\}}\right)\mathbb{E}|\Delta^m|^2 + \frac{4h^3}{\mu}\sum_{i=1}^d \frac{(L_i^3 + H_i^2)}{\phi_i^2}\,. \tag{88}$$

Since $h < \frac{\mu\min\{\phi_i\}}{8L^2}$, we obtain (68). $\qquad\square$

# D  PROOF OF PROPOSITION 4.2

*Proof of Proposition 4.2.* For this special target distribution $p$, the objective function is $f(x) = \sum_{i=1}^d \frac{|x_i|^2}{2}$. With $\alpha = 0$ and $\phi_i = 1/d$, we have: $x_i^{m+1} = x_i^m$ for all $i \ne r^m$ and

$$x_{r^m}^{m+1} = (1 - dh)x_{r^m}^m + \sqrt{2dh}\xi^m\,.$$

Therefore for all $i = 1, 2, \ldots, d$, we have

$$\mathbb{E}|x_i^{m+1}|^2 = \frac{1}{d}\mathbb{E}\left(|x_i^{m+1}|^2\big|r^m = i\right) + \left(1 - \frac{1}{d}\right)\mathbb{E}\left(|x_i^{m+1}|^2 \mid r^m \ne i\right)$$
$$= \frac{1}{d}\mathbb{E}\left(|(1 - dh)x_i^m + \sqrt{2dh}\xi^m|^2\big|r^m = i\right) + \left(1 - \frac{1}{d}\right)\mathbb{E}\left(|x_i^m|^2\right)$$
$$= \left(1 - 2h + dh^2\right)\mathbb{E}|x_i^m|^2 + 2h \tag{89}$$

where we use $\mathbb{E}_\xi\left|x_i^m - dhx_i^m + \sqrt{2dh}\xi^m\right|^2 = (1 - dh)^2|x_i^m|^2 + 2dh$ in the last equation. By summing (89) over $i$, we obtain

$$\mathbb{E}|x^{m+1}|^2 = \left(1 - 2h + dh^2\right)\mathbb{E}|x^m|^2 + 2dh\,.$$

Using it iteratively, and considering $\mathbb{E}|x^0|^2 = 3d$, we have:

$$\mathbb{E}|x^m|^2 \ge 3d\left(1 - 2h + dh^2\right)^m + \left(1 - \left(1 - 2h + dh^2\right)^m\right)\frac{2dh}{2h - dh^2}$$
$$= d\left(1 - 2h + dh^2\right)^m + \frac{2d}{2 - dh} + 2d\left(1 - \frac{1}{2 - dh}\right)\left(1 - 2h + dh^2\right)^m$$
$$\ge d\left(1 - 2h\right)^m + \frac{2d}{2 - dh}\,,$$

where we use $dh \leq 1$ in the last inequality.

Since

$$W(q_m, p) \geq \left( \int |x|^2 q_m(x) \, \mathrm{d}x \right)^{1/2} - \left( \int |x|^2 p(x) \, \mathrm{d}x \right)^{1/2} = \left( \int |x|^2 q_m(x) \, \mathrm{d}x \right)^{1/2} - \sqrt{d} \,,$$

we have

$$
\begin{aligned}
W(q_m, p) &\geq \left( \int |x|^2 q_m(x) \, \mathrm{d}x \right)^{1/2} - \sqrt{d} \geq \frac{d \left(1 - 2h\right)^m + \frac{2d}{2 - dh} - d}{\sqrt{d \left(1 - 2h\right)^m + \frac{2d}{2 - dh}} + \sqrt{d}} \\
&\geq \frac{\sqrt{d}}{3} \left(1 - 2h\right)^m + \frac{d^{3/2} h}{6} \\
&\geq \exp\left(-2mh\right) \frac{\sqrt{d}}{3} + \frac{d^{3/2} h}{6} \,,
\end{aligned}
$$

where in the last inequality we use

$$\sqrt{d \left(1 - 2h\right)^m + \frac{2d}{2 - dh}} + \sqrt{d} \leq 3\sqrt{d}.$$

Therefore, we finally prove (26). $\qquad\square$

