# OpenReview forum: "Random Coordinate Langevin Monte Carlo"
_ICLR.cc/2021/Conference — Reject_

### Official Review · AnonReviewer4 · 2020-10-19
**All the theoretical results of the paper are inetersting and worth being published. However, the proofs as presented in the current version are flawed and require non negligible additional technical arguments.**

**Rating:** 6
**Confidence:** 5

**Review:**

$\textbf{Post rebuttal update}$

I am still positive about the paper and believe that it deserves to be published. However, I agree with Reviewer 2 that some non-trivial rewriting is necessary. The flaws are most likely not very hard to be repaired but it will require substantial additional work and the new version needs to be carefully checked. This is the main reason why I downgraded my score from 7 to 6.
*****************

$\textbf{Description of the contributions}$

The paper studies the problem of sampling from a smooth multivariate density defined via a strongly
convex potential function. The main focus is on the analysis of a version of the Langevin Monte Carlo
algorithm in which the computation of the full gradient at each iteration is replaced by the computation
of a randomly selected partial derivative. The main results of the paper are theoretical guarantees
formulated in terms of the Wasserstein distance between the sampling distribution and the target
distribution.

$\textbf{Evaluation}$

I find the results very interesting and worth being published. However, the current version of the paper
suffers from some shortcomings that have to be repaired. The most important shortcomings are listed below,
the others are provided as specific remarks.

P1. The most important problem is the way the proof of $\textbf{Prop A.1}$ is written. I believe the result is true, however
the arguments provided in the proof should be clarified (Please refer to point 8vin the list below). In particular,
it is more appropriate to refer to the conditional distribution of $X_{r^m}(t)$ given all the other coordinates, rather
than to the marginal distribution.

P2. The second shortcoming, slightly less important, is that the paper contains no concrete example of potential
for which the results of this paper (for instance, $\textbf{Theorem 4.2}$) improves on the previously available results.

P3. An important part of the supplementary material is devoted to the proof of the second claim of $\textbf{Theorem 4.1}$.
My impression is that this second claim, which provides exponential ergodicity of the Markov chain, is of little
interest and does not really fit the framework of the paper. Indeed, while all the remaining results come with
explicit dependence of the constants on the dimension and the condition number, the constants involved in (20)
are not specified at all. My suggestion is to remove the second claim of $\textbf{Theorem 4.1}$ (starting from "Furthermore,
if ... ").

$\textbf{Specific remarks}$

1. Page 2: "d-dimensional Brownian motion with independent components" a Brownian motion has always independent components, no need to additionally underline this property.
2. Page 2: "under mild conditions, the SDE converges exponentially fast to the target distribution" First, conditions are not really mild. Second, it is not the SDE that converges, but the distribution of its solution at time t.
3. Page 2: The sentence "One main drawback of LMC is that its dependence on the problem dimension d is rather bad." is too strong. The dependence in the strongly convex case is linear in d, which can hardly be qualified as "rather bad".
4. Page 2: "with LMC depends strongly" -> "with LMC depend strongly"
5. Page 2: the authors claim that "RC-LMC [...] is cheaper than the classical LMC" This claim should be carefully reformulated. What the results of this paper imply, under the most favorable interpretation, is that the obtained result for RC-LMC is better than the best known result for the LMC. It might very well happen that the result for the LMC is improvable and that the LMC is as fast as the RC-LMC.
6. Page 4: "When we compare (5) with the classical LMC (1), we see that in the updating formula, the gradient
is replaced by a partial derivative in a random direction $r_m$" I am not sure that this claim is true. It would be correct if the authors were adding the Gaussian noise term to all the coordinates, but this is not done in the current definition.
On a related note, in equation (7), the dimension d of the Brownian motion does not match with dimension 1 of the left-hand side.
7. Section 3: It might be that I missed something, if not, it would be useful to highlight that the notation $|v|$ is used for the Euclidean norm of $v$.
8. The proof of Theorem 4.1 is not valid. Indeed, the fact that $p(x)$ is the invariant distribution of the Markov chain is not correctly proved. Even if we admit that the (questionable) argument with the EDS is correct, it only proves that the marginal distribution
of one coordinate and the joint distribution of the remaining coordinates coincide with those of $p$. This does not imply that the entire distribution coincides with that of $p$.
Another flaw in the proof of the theorem concerns line 8 of the proof. This inequality should further justified. I see very well how to prove it in the case when the Markov kernel is absolutely continuous wrt to the Lebesgue measure. However, I am not quite sure that the Markov kernel used in this theorem is indeed abs-cont.

---

> ### Author Response · Authors · 2020-11-17
> **Response to the reviewer**
>
> We thank the reviewer for valuable suggestions! We have revised our paper accordingly. We provide the following response to the reviewer's three remarks.
>
> R1: The most important problem is the way the proof of Prop A.1 is written. I believe the result is true, however the arguments provided in the proof should be clarified. In particular, it is more appropriate to refer to the conditional distribution of $X_{r^m}(t)$ given all the other coordinates, rather than to the marginal distribution.
>
> A1: Given the reference to conditional distribution, we believe that the reviewer meant Theorem 4.1 (Proposition 4.1 in the revision) rather than of Proposition A.1 (Proposition A.2 now). We have revised the proof accordingly and tried to make it more precise, as seen in the current version in Appendix A.
>
> R2: The second shortcoming, slightly less important, is that the paper contains no concrete example of potential for which the results of this paper (for instance, Theorem 4.1) improves on the previously available results.
>
> A2: We thank the referee for this suggestion. When the Hessians are Lipschitz (Case 2 in our paper), RC-LMC always outperforms LMC by a factor of $d^{1/2}$. Since this assumption is quite weak, examples that yield such improvement are abundant. Case 1 concerns case of Lipschitz-continuous gradients, and our study suggests that improvement over LMC can be expected when $d^2\kappa\geq \left(\sum^d_{i=1}\kappa_i\right)^2$. Such is the case for the following example: $f(x)=dx^2_1+\sum^d_{i=2} x^2_i$,where $\kappa=\kappa_1=d$ but $\kappa_i=1$ for $i\neq 2$. We see that $d^2\kappa=d^3\gg 4d^2\geq (\sum \kappa_i)^2$ when $d$ is large, so RC-LMC can be expected to improve over LMC in this case.
>  We have added Remarks 4.1 and 4.2 in the revised paper to explain these points.
>
>
> R3:  An important part of the supplementary material is devoted to the proof of the second claim of Theorem 4.1. My impression is that this second claim, which provides exponential ergodicity of the Markov chain, is of little interest and does not really fit the framework of the paper. Indeed, while all the remaining results come with explicit dependence of the constants on the dimension and the condition number, the constants involved in (20) are not specified at all. My suggestion is to remove the second claim of Theorem 4.1 (starting from "Furthermore, if ... ").
>
> A3: We agree with the reviewer. The dependence of $r$ and $R$ is not explicit, making the result not compatible with the rest in the paper. We have  revised the theorem statement (it is now called Proposition 4.1), and moved to the result on exponential convergence to Proposition A.1 in the Appendix.

---

> > ### Author Response · Authors · 2020-11-24
> > **More responses**
> >
> > All other specific comments have been addressed in the revised paper as well. Here we list the significant changes.
> >
> > R4: The proof of Theorem 4.1 is not valid. Indeed, the fact that $p(x)$ is the invariant distribution of the Markov chain is not correctly proved. Even if we admit that the (questionable) argument with the EDS is correct, it only proves that the marginal distribution of one coordinate and the joint distribution of the remaining coordinates coincide with those of $p$. This does not imply that the entire distribution coincides with that of $p$. Another flaw in the proof of the theorem concerns lines 8 of the proof. This inequality should further justified. I see very well how to prove it in the case when the Markov kernel is absolutely continuous wrt to the Lebesgue measure. However, I am not quite sure that the Markov kernel used in this theorem is indeed abs-cont.
> >
> > A4: We thank the reviewer for pointing this out! We agree with the reviewer that the previous proof contains flaws. We have corrected the proof in the revision. Since this theorem is not compatible with the other results on non-asymptotic convergence rates, we have moved it to Appendix (now named  Proposition A.1 in the revision).
> >
> > In the new proof, we show that the conditional distribution of $X^m_i$ (with other components fixed) is preserved, as seen in the proof of Proposition A.1 (31). For the original inequality in line 8 of the proof, we revised and put it between (31) and (32). Moreover, in the revision, we consider the distribution of $X^m$ instead of its density. Therefore, we no longer need the Markov kernel to be absolutely continuous.
> >
> > R5: Page 2: the authors claim that ``RC-LMC [...] is cheaper than the classical LMC" This claim should be carefully reformulated. What the results of this paper imply, under the most favorable interpretation, is that the obtained result for RC-LMC is better than the best-known result for the LMC. It might very well happen that the result for the LMC is improvable and that the LMC is as fast as the RC-LMC.
> >
> > A5: We agree with the reviewer. We have revised our statement to make it more precise.
> >
> > R6: Page 4: ``When we compare (5) with the classical LMC (1), we see that in the updating formula, the gradient is replaced by a partial derivative in a random direction $r_m$" I am not sure that this claim is true. It would be correct if the authors were adding the Gaussian noise term to all the coordinates, but this is not done in the current definition. On a related note, in equation (7), the dimension d of the Brownian motion does not match with dimension 1 of the left-hand side.
> >
> > A6: We agree with the reviewer. This is a typo. We fixed it by using $B_{t,d}$ to represent $d$-dimensional Brownian motion (as see in (2)). Now, $B_t$ represents $1$-dimensional Brownian motion (as see in (7)).

---

> > > ### Author Response · Authors · 2020-11-25
> > > **Thanks for the further comments**
> > >
> > > We agree completely that more experiments in more practical settings are needed. This is our next research goal. In this paper, we focus on theoretical results. We run the experiments to verify the theoretical predictions. We believe this verification serves as the foundation for further investigation in practical settings

---

### Official Review · AnonReviewer1 · 2020-10-27
**Interesting upper and lower bounds of Random Coordinate LMC under various smoothness assumptions**

**Rating:** 6
**Confidence:** 4

**Review:**

Post rebuttal update:
I read the other reviewers' responses, and, although I am still positive about this paper, I agree with R2 and R4 that safely fixing the theoretical proofs would require a full revision. For this reason, I am lowering my score to 6.

%%%%%%%%%%%%%%%%%%%%%%%%%

The authors propose a variant of Unadjusted Langevin Algorithm by replacing the full gradient of the log-density by the gradient of a single coordinate selected at random according to some chosen probability distribution phi. When the log-density of the target distribution is gradient Lipschity and strongly convex, and the step size for updating a coordinate is inversly proportional to the probability of selecting it, the authors show approximate convergence in 2-Wasserstein distance of Random Coordinate LMC (RC-LMC) to the target distribution. The convergence guarantees, in terms of cost, match the ones of classical LMC in terms of dimension and accuracy dependence.

In the case where all dimensional Lipschitz constants are known, the authors propose a new choice of coordinate selection probability distribution phi and step sizes that yield similar convergence guarantees, but with d^2 kappa dependence replaced by (sum_i kappa_i) where kappa is the global condition number, and kappa_i's are the dimensional condition numbers. Hence, this yields improved convergence guarantees in the case of high dimensional and highly skewed log-density.

In the case where, in addition, the log-density has a Lipschitz Hessian, and under a proper choice of the coordinate selection distribution phi and step sizes depending on the dimensional gradient and Hessian Lipschitz constants, the authors show convergence of RC-LMC in 2-Wasserstein distance with rate O(d^3/2 epsilon), improving upon the best known rate for this setting.

The authors show a lower bound for RC-LMC in the gradient and Hessian Lipschitz case, matching the previously shown upper bound. Such a lower bound is appreciated, especially in the literature of Langevin dynamics based sampling algorithm where convergence upper bounds are plentiful and not much is known about lower bounds (especially in the deterministic gradient setting).

Finally, the authors perform a numerical experiment in which they estimate the expectation of some test function of some randome variable following a skewed Gaussian distribution from N samples. They demonstrate that RC-LMC, with our without the knowledge of the dimensional Lipschitz constant, converge faster than classical LMC.

Concerning this experiment, the various methods converge to different saturation thresholds. However, since the objective is strongly log-concave, all methods should converge arbitratily close to the target distribution when choosing the step size small enough (or using a decaying step size). The only limitant factor should then only be due to N being finite, which is common to all methods. Could you please mention and argue upon the choice of step sizes for each method? It would also be nice to plot the optimal saturation threshold for estimating the expecation of the test function from N samples, which should be computable exactly for a Gausian target distribution.

---

> ### Author Response · Authors · 2020-11-17
> **Response to the reviewer**
>
> We thank the reviewer for the thorough evaluation and positive feedback! Our response to the reviewer's comments is as follows.
>
> R1: Concerning this experiment, the various methods converge to different saturation thresholds. However, since the objective is strongly log-concave, all methods should converge arbitrarily close to the target distribution when choosing the step size small enough (or using a decaying step size). The only limiting factor should then only be due to $N$ being finite, which is common to all methods. Could you please mention and argue upon the choice of step sizes for each method? It would also be nice to plot the optimal saturation threshold for estimating the expectation of the test function from N samples, which should be computable exactly for a Gaussian target distribution.
>
> A1: We agree with the reviewer that dependence on $N$  is important. In the numerical experiment we choose $N$ large enough to suppress the central limit theorem error of ${1}/{\sqrt{N}}$ and truly measure the distance between $q^m$ and $p$, as the main purpose is to demonstrate the accuracy and effectiveness of different algorithms in terms of dependence of condition number and dimension.
>
> We also agree that the choice of $h$ is a very delicate issue. Indeed, with smaller time stepsize $h$, the saturation error (called error plateau in the paper) will become smaller. However, smaller $h$ yields slower error decay and hence higher total cost. To make for a richer comparison,  we now test multiple choices of $h$ for LMC in our numerical tests reported in Figure 1. All these choices of $h$ for LMC, however, give results that are worse in comparison to RC-LMC.

---

### Official Review · AnonReviewer2 · 2020-10-28
**A nice algorithm! Analysis may have some issue.**

**Rating:** 4
**Confidence:** 4

**Review:**

This paper studies Langevin Monte Carlo (LMC) in the high dimensional regime. To reduce the computational cost, the authors proposed the Random Coordinate LMC (RC-LMC) algorithm that only updates one of its coordinates randomly at each iteration. Despite the fact that only one coordinate is updated, the authors prove that RC-LMC still converges fast to the stationary distribution. Given some nice properties of the log-density functions, the total cost of RC-LMC could be smaller than LMC especially when the function is highly skewed in a high dimension space. This paper is well written and clearly presented. The experiment is somewhat limited since the dimension of the problem is not high to me. It would also be nice if we could see how the proposed algorithm performs on real datasets of Beyesian sampling problems. I also have some comments on the technical analysis of this paper as follows.

The non-asymptotic analysis of LMC is also studied in Xu et al. (2018), which develops a quite different analysis from other works by directly showing the ergodicity of the Markov chain generated by LMC. It seems that the analyses in that paper and the current submission are closely related.

Xu P, Chen J, Zou D, Gu Q. Global convergence of Langevin dynamics based algorithms for nonconvex optimization. In Advances in Neural Information Processing Systems 2018 (pp. 3122-3133).

The cost in Theorem 4.2 depends on the sampling probability for choosing the coordinate. It should be noted that if some \phi_i is significantly smaller than 1/d, the total cost may have a worse dependence on dimension.

In the proof of the ergodicity of the Markov chain X^m, is it true that both r and R are constants independent of the problem dimension?

In Theorem 4.1 and Proposition A.1, the authors claim that the stationary distribution of Markov Chain X^m is p(x). I am not sure whether this is correct. Note that p(x) is the stationary distribution of the Markov chain X_t defined by the SDE in (2). We know that X^m is a discretized chain based on X_t. Therefore, as long as the step size is not zero, there will be some discrepancy between these two Markov chains. Thus their stationary distribution could not be the same.

---

> ### Author Response · Authors · 2020-11-17
> **Response to the reviewer**
>
> We thank the reviewer for providing valuable suggestions, and we are encouraged that the reviewer likes the algorithm. The reviewer's biggest concerns are about the analytical result, as seen in Remark 4 and 5 (R4, R5) below. We believe these concerns can be addressed; they are due in part to our possibly confusing notations.
> Below we list our answers to all remarks of the reviewer.
>
> R1: It would be nice if we could see how the proposed algorithm performs on real datasets of Bayesian sampling problems.
>
> A1: We  agree completely that more experiments in more practical settings are needed. This is our next research goal. The current paper is theoretical, and our experiments are intended simply to verify the theoretical predictions --- which they do. We believe this verification serves as the foundation for further investigation in practical settings.
>
> R2:  The non-asymptotic analysis of LMC is also studied in Xu et al. (2018), which develops a quite different analysis from other works by directly showing the ergodicity of the Markov chain generated by LMC. It seems that the analyses in that paper and the current submission are closely related.
>
> A2: We thank the reviewer for pointing out this reference. Indeed, Xu et al (2018) established the ergodicity of the LMC Markov chain using an argument based on a Lyapunov function. This approach is similar in spirit to our results in Proposition 4.1 and Proposition A.1, where we show the Markov chain property of the SDE (the continuous version of RC-LMC). However, our strategy for the non-asymptotic results (presented in Section 4.2 and 4.3) is quite different --- we use a coupling approach motivated by Dalalyan and Karagulyan (2019).
>     We agree that the paper of Xu et al is closely related to ours, so we have added it to our reference list and provided some brief comments in Remark A.1.
>
> R3: The cost in Theorem 4.2 depends on the sampling probability for choosing the coordinate. It should be noted that if some $\phi_i$ is significantly smaller than $1/d$, the total cost may have a worse dependence on dimension.
>
> A3: Good point! According to Theorem 4.2 (which has become Theorem 4.1 in the revised version), RC-LMC might have a worse dependence on dimension if $\phi_i$ is chosen arbitrarily. This is not surprising: If the sampling strategy runs somehow contrary to the stiffness structure of $f$, then the algorithm essentially wastes too much computational cost to address unimportant directions. But this situation will not arise in a reasonable implementation: If the stiffness structure of $f$ is unknown, one should merely choose $\phi_i=1/d$. Already, RC-LMC outperforms vanilla LMC for this choice of weights, both for Lipschitz-continuous gradient and Lipschitz-continuous Hessian. Our revised paper clarifies this point, and includes detailed discussions in in Remarks 4.1 and 4.2.
>
>
> R4: In the proof of the ergodicity of the Markov chain $X^m$, is it true that both $r$ and $R$ are constants independent of the problem dimension?
>
> A4: Good question. According to a paper in our reference list, by Mattingly, J., Stuart, A., and Higham, D. (2002), both $r$ and $R$ should depend on $d$. So indeed this result, though providing some intuition about the convergence, is significantly weaker than the non-asymptotic convergence results that we show in later subsections. However, this is a result regarding the continuum limit of the algorithm --- the SDE --- not the algorithm itself. Our non-asymptotic analysis for the algorithm itself does come with explicit dependence on all parameters, as shown in Section 4.2 and 4.3. In the revision, we have moved the part of the result concerning the continuous SDE to the appendix,  to avoid confusion and to shed more light on our main results. We have also added Remark A.1 in the revision, as a response to the reviewer's suggestion.
>
> R5: In Theorem 4.1 and Proposition A.1, the authors claim that the stationary distribution of Markov Chain $X^m$ is $p(x)$. I am not sure whether this is correct. Note that $p(x)$ is the stationary distribution of the Markov chain $X_t$ defined by the SDE in (2). We know that $X^m$ is a discretized chain based on $X_t$. Therefore, as long as the step size is not zero, there will be some discrepancy between these two Markov chains. Thus their stationary distribution could not be the same.
>
> A5: This is a very important point, and we believe there is a misunderstanding here. In our paper $X^m = X(T^m)$, so it is a skeleton of the continuous SDE, but not $x^m$ whose trajectory of course contains some discretization error compared with the SDE. The notational similarity between $X^m$ and $x^m$ and possibly led to a misunderstanding.
>
> The stationary distribution for $x^m$ is indeed not $p(x)$, exactly as what the reviewer suggests, but the stationary distribution for $X^m$ is in fact $p(x)$. The proof is given in Appendix A, where we need to analyze the SDE (7), instead of its discrete version (5).

---

### Official Review · AnonReviewer3 · 2020-10-29
**Review of "Random Coordinate Langevin Monte Carlo"**

**Rating:** 4
**Confidence:** 4

**Review:**

This paper generalizes the Langevin within Gibbs sampler to be able to put different frequencies over different coordinates. The idea is cute. The result, however, is not convincingly better than the vanilla Langevin algorithm.

The convergence rate for the current method for strongly convex and Lipschitz smooth case scales as O(d^2/\epsilon^2) in Wasserstein 2 distance, where every step requires partial derivative in one coordinate. This is to be compared to O(d/\epsilon^2) gradient computations required for the vanilla Langevin algorithm to converge. Although in terms of number of partial derivatives, they are comparable to each other, current computation infrastructure has made gradient computation a lot cheaper than d number of sequential  partial derivative computations.

One possible use case, as the authors mentioned, might be that certain dimensions are more stiff than the others, calling for more careful exploration. In practice, however, the stiff dimensions change with the state and it is challenging to detect these stiff directions on the fly.

#################################################################################
TLDR

Pros:
Generalizes Langevin within Gibbs and achieves convergence guarantees.

Cons:
Not outperforming (sometimes even underperforming) current methods.

Related work:
Please also check the related work: MALA-within-Gibbs samplers for high-dimensional distributions with sparse conditional structure, X. T. Tong, M. Morzfeld, Y. M. Marzouk, 2019.

---

> ### Author Response · Authors · 2020-11-17
> **Response to the reviewer**
>
> We thank the reviewer for the suggestions. Based on our reading of the reviewer's comments, we believe there are some misunderstandings, potentially because the introduction of the paper did not stress enough some important factors. We have revised the manuscript in a way that (we hope) clarifies matters for the reviewer and addresses the concerns.
>
> We stress four main points:
>
> 1. Regarding the performance of the method: We do not agree with the reviewer's comment stating that the new method (RC-LMC) does not outperform the current method.
> In fact, in both cases that we study (Lipschitz-continuous gradient and Lipschitz-continuous Hessian), the new method (RC-LMC) outperforms the vanilla LMC. In the former case, a certain stiffness structure is needed to obtain improvement, whereas in the latter case, RC-LMC reduces the cost by  $d^{1/2}$ , regardless of the stiffness structure of the problem.
> We believe the latter case reflects a very strong advantage of RC-LMC over vanilla LMC, which is not seen in comparing RCD with GD. This property seems to be unique to the sampling setting.
> We have revised the introduction to stress these advantages towards the end and further discussed in detail the comparison in Remark 4.1 and 4.2 for the two settings.
>
> 2. Regarding the cost of computing the gradient: The reviewer also commented on the cost of computing the gradient vs. computing partial derivatives. We agree that there are certainly models in which the whole gradient can be computed efficiently using appropriate techniques, such as automatic differentiation, as we mention in the paper. In such cases, the advantage of RC-LMC is not evident. However, there are also many interesting applications in which the cost of calculating the gradient is indeed approximately $d$ times the cost of computing one partial derivative. ERM (empirical risk minimization) problems (not discussed in the paper) and PDE-based inverse problems (we listed a few examples in the paper) are in this category. The graph example mentioned in the paper is another such illustration.
> We are well aware of this consideration about coordinate descent, which arises also in the optimization setting.
> Various forms of CD remain popular in optimization and machine learning because of their practicality on problems such as those mentioned in the previous paragraph. In Bayesian sampling, Gibbs sampling (which can be viewed as a coordinate algorithm) also  relies on per-dimension updates being cheap. The wide popularity of these methods indicate the potential applications of the method proposed in our work.
>
> 3. Regarding the stiffness structure: It seems that the reviewer believes the prior knowledge of the stiffness structure is needed for RC-LMC to outperform LMC.  In fact, RC-LMC can outperform LMC even without detailed knowledge of Lipschitz constants. The uniform-sampling choice $\phi_i = 1/d$ already gives RC-LMC an advantage, in terms of complexity.  If Lipschitz constants are known, the optimal sampling strategy yields an even greater advantage for RC-LMC.  We discuss these points in Remarks 4.1 and 4.2.
>
> 4. Regarding the paper by Tong-Morzfeld-Marzouk: We thank the reviewer for mentioning the reference. The setting of their work is quite different from ours.
>   1 In (X.T. Tong), the algorithm updates coordinates in a specific order, not randomly. As a result, the analysis of the algorithms are also quite different. (The same is true in the optimization setting, where the analysis of cyclic CD is quite different from analysis of randomized CD.)
>    2 The assumption in (X.T. Tong) is very different from ours as well. In their paper, $f$ is of low rank, strongly convex, with the gradient $\nabla f$ bounded. Our work considers a more classical setting in sampling literature (for example when classical LMC is analyzed): $f$ strongly convex with Lipschitz gradient $\nabla f$.
> There is certainly a relationship with this paper, so we  have added the reference and some corresponding discussion at the end of Section 1.
>
> In summary, Random coordinate descent (RCD) is a very popular optimization method, and it is shown to outperform GD (Wright, S. J. (2015) in the reference list) in some regimes. Our work  serves as a parallel discussion in the sampling setup. To our great surprise, the advantage of RC-LMC over LMC is significantly stronger than that of RCD over GD. When Hessian is Lipschitz, RC-LMC  wins by a factor of at least $d^{1/2}$.
>
> We hope that our explanations and revisions have addressed the referee's concerns.

---

### Decision · Program_Chairs · 2021-01-07
**Final Decision**

**Decision:**

Reject

**Comment:**

This paper proposes a new sampling method named Random Coordinate LMC (RC-LMC), which integrates the idea of randomized coordinate descent and Langenvine dynamic. The authors prove the total complexity of RC-LMC for log-concave probability distributions, which are better than that of LMC under different settings. The idea of this paper is very neat and the reviewers are in general positive about it. However, as pointed out by one of the reviewers and seconded by the other reviewers, the proof in the original submission is flawed, and the fix needs some substantial work. The new version needs to be carefully checked before publication, which is far beyond the review process of ICLR. Therefore, I encourage the authors to carefully revise the paper and submit it to the next conference.